



# 1    Investigation of global nitrate from the AeroCom Phase III experiment

Huisheng Bian[1,2], Mian Chin[2], Didier A. Hauglustaine[3], Michael Schulz[4], Gunnar Myhre[5,6],
Susanne E. Bauer[7,8], Marianne T. Lund[6], Vlassis A. Karydis[9], Tom L. Kucsera[10], Xiaohua Pan[11],
Andrea Pozzer[9], Ragnhild B. Skeie[6], Stephen D. Steenrod[10], Kengo Sudo[12], Kostas
Tsigaridis[7,8], Alexandra P. Tsimpidi[9], and Svetlana G. Tsyro[4]
[1] Joint Center for Environmental Technology UMBC, Baltimore, MD, USA
[2] Laboratory for Atmospheres, NASA Goddard Space Flight Center, Greenbelt, MD, USA
[3] Laboratoire des Sciences du Climat et de l'Environnement (LSCE), UMR8212, CEA-CNRS-UVSQ, Gif-
sur-Yvette, France
[4] Norwegian Meteorological Institute, Blindern, Norway
[5] Department of Geosciences, University of Oslo, Oslo, Norway
[6] Center for International Climate and Environmental Research-Oslo, Oslo, Norway
[7] The Earth Institute, Center for Climate Systems Research, Columbia University, New York, USA
[8] NASA Goddard Institute for Space Studies, New York, USA
[9] Max Planck Institute for Chemistry, 55128 Mainz, Germany
[10] Universities Space Research Association, GESTAR, Columbia, MD, USA
[11] School of Computer, Mathematical and Natural Sciences, Morgan State University, Baltimore, MD, USA
[12] Center for Climate System Research, University of Tokyo, Tokyo, Japan.

## 22    Abstract

An assessment of global nitrate and ammonium aerosol based on simulations from nine
models participating in the AeroCom Phase III study is presented. A budget analyses was
conducted to understand the typical magnitude, distribution, and diversity of the aerosols
and their precursors among the models. To gain confidence on model performance, the
model results were evaluated with various observations globally, including ground station
measurements over North America, Europe, and East Asia for tracer concentrations and
dry and wet depositions, as well as with aircraft measurements in the Northern
Hemisphere mid-high latitudes for tracer vertical distributions. Given the unique
chemical and physical features of the nitrate occurrence, we further investigated the
similarity and differentiation among the models by examining: 1) the pH-dependent $NH_3$
wet deposition; 2) the nitrate formation via heterogeneous chemistry on the surface of
dust and sea-salt particles; and 3) the nitrate coarse mode fraction (i.e., coarse/total). It is
found that $HNO_3$, which is simulated explicitly based on full $O_3$-$HO_x$-$NO_x$-aerosol
chemistry by all models, differs by up to a factor of 9 among the models in its global
tropospheric burden. This partially contributes to a large difference in $NO_3^-$, whose
atmospheric burden differs by up to a factor of 13. Analyses at the process level show
that the large diversity in atmospheric burdens of $NO_3^-$, $NH_3$, and $NH_4^+$ is also related to
deposition processes. Wet deposition seems to be the dominant process in determining
the diversity in $NH_3$ and $NH_4^+$ lifetimes. It is critical to correctly account for contributions
of heterogeneous chemical production of nitrate on dust and sea-salt, because this process
overwhelmingly controls atmospheric nitrate production (typically >80%) and determines
the coarse and fine mode distribution of nitrate aerosol.

## 46    1. Introduction

Atmospheric aerosols adversely affect human health and play an important role in
changing the Earth's climate. A series of multimodel studies have been coordinated by
the international activity of Aerosol Comparisons between Observations and Models





(AeroCom) in its Phase I and II model experiments that have systematically assessed the
presence and influence of almost all major atmospheric anthropogenic and natural
aerosols (such as sulfate, dust, and carbonaceous aerosols) (e.g., Kinne et al., 2006;
Schulz et al., 2006; Textor et al., 2006; Koch et al., 2009; Huneeus et al., 2011; Tsigaridis
et al., 2014; Kim et al., 2015). Very little attention has been drawn to nitrate aerosol other
than its contribution to radiative forcing (Myhre et al., 2013). One obvious reason is that
not many models used to include nitrate owing to the chemical complexity of nitrate
formation. However, atmospheric nitrate aerosol not only exerts direct effects on air
quality and climate, but also uniquely impacts the Earth system by being directly
involved in tropospheric chemistry and constraining net primary productivity, hence
altering carbon sequestration and ecological effects, via its deposition (Prentice et al.,
61     2001).
Atmospheric nitrate contributes notably to total aerosol mass in the present-day,
especially in urban areas and agriculture regions. Nitrate is about a quarter of sulfate in
terms of overall global burden, AOD, and direct forcing at the present-day according to
the study of AeroCom II direct forcing experiment (Myhre et al., 2013). This conclusion
is confirmed by recent publications using various individual models and emission
inventories (Bellouin et al; 2011; Bauer et al., 2007; Hauglustaine 2014; Karydis et al.,
2016; Mezuman et al., 2016; Paulot et al., 2016). Regionally, considerable evidences
from in-situ measurements (Bessagnet et al., 2014; Haywood et al., 2008; Jimenez et al.,
2009; Malm et al., 1994; Vieno et al., 2016) and model results (Karydis et al., 2011;
Ensberg et al., 2013; Trump et al., 2015) indicate that nitrate becomes one of the major
aerosol species in urban and agriculture environments. For example, nitrate concentration
is about half of sulfate during the summer season in Beijing (Zhou et al., 2016) and
represents a large portion of wintertime aerosol mass in the San Joaquin Valley in
California (Pusede et al., 2016).
More importantly, the importance of aerosol nitrate is likely to increase over the century
with a projected decline in $SO_2$ and $NO_x$ emissions and increase in $NH_3$ emissions (IPCC,
2013). With the reduction of $SO_2$ emissions, less atmospheric $NH_3$ is required to
neutralize the strong acid $H_2SO_4$. The excess of $NH_3$ results in gaseous $HNO_3$ and $NH_3$
entering the condensed phase, and their subsequent dissociation yields nitrate and
ammonium ions. The trend of future nitrate depends on which is the limited species, $NO_x$
or $NH_3$, for nitrate formation (Tsimpidi et al., 2007; 2008). Generally, our atmosphere, at
its current and foreseeable near future, is still in an $NH_3$-limited condition according to
sensitivity studies by Heald et al. (2012) and Walker et al. (2012). Almost all global
models predicted an overall increase of atmospheric nitrate burden during this century
based on current available emission inventories (Bauer et al 2007; 2016; Bellouin et al.,
2011; Hauglustaine et al., 2014; Li et al., 2014). For example, using CMIP5 future
emission projections, Bellouin et al. (2011) concluded that, by 2090, nitrate would
become an important aerosol species in Europe and Asia, contributing up to two thirds of
the globally averaged anthropogenic optical depth. However, the predicted trend of
surface nitrate is mixed. Some studies estimated a consistent increase of surface nitrate
(Bellouin et al., 2011), while others pointed out that this increase might vanish or even
reverse over some regional urban areas due to the decline of $NO_x$ emissions (Bauer et al.,





2016; Pusede et al., 2016; Trail et al., 2014). Nevertheless, the potentially increasing
importance of nitrate in climate and its large uncertainty in future surface nitrate
predictions urge us to characterize model performance and understand the
physicochemical mechanisms behind the diversity of nitrate simulations.
Nitrate is also important in that its formation directly affects tropospheric chemistry.
First, the formation of nitrate, through either aqueous phase chemical reaction between
$HNO_3$ and $NH_3$ (Metzger et al., 2002; Kim et al., 1993) or heterogeneous reaction of
nitrogen species such as $HNO_3$, $NO_3$, and $N_2O_5$ on the surface of dust and sea salt aerosol
particles (Bauer et al., 2004; 2005; Bian et al., 2003; Dentener 1996; Liao et al., 2003),
converts gas phase nitrogen species into aerosols. Consequently, the global tropospheric
$NO_x$ concentration and the rate of conversion of $N_2O_5$ to $HNO_3$ will be reduced (Riemer
et al., 2003), which in turn leads to the reduction of atmospheric oxidants. For example,
global tropospheric $O_3$ can be reduced by 5% (Bauer et al., 2007) and tropical Atlantic
OH by 10% (Bian et al., 2003) just through the heterogeneous reactions of nitrogen
radicals on dust. Second, the most important removal path for nitrogen from the
atmosphere is the formation of $HNO_3$, which is subsequently deposited (Riemer et al.,
2003). Since $HNO_3$ is subject to partitioning between the gas and aerosol phases, the
lifetimes of nitrogen species can be shortened by the formation of tropospheric nitrate
aerosol because the loss of total $HNO_3$ will be accelerated by a much higher dry
deposition in the aerosol phase.
Large nitrogen deposition occurs over both land and ocean (Dentener et al., 2006;
Kanakidou et al., 2012; 2016). Nitrogen deposition can either benefit or impair ecosystem
productivity depending on the initial balance of nutrients since different ecosystems have
different Nr (reactive nitrogen) availability and retention (Galloway et al., 2004; Prentice
et al., 2001). If fixed Nr is deposited as nitrate in forests, it may act as a "fertilizer,"
stimulating growth and thus enhancing carbon sequestration (Fowler et al., 2015). But
when the accumulated deposition exceeds the nutritional needs of the ecosystem, nitrogen
saturation may result (Fenn et al., 1996). Soil fertility declines due to the leeching of
cations (Milegroet and Cole, 1984) and, thus, carbon uptake diminishes. The balance
between fertilization and saturation depends on the spatial and temporal extent of
nitrogen deposition. In order to determine the extent to which the emissions of air
pollutants will have to be reduced and whether the environment needs to be protected
from damage, it is essential to know where and by how much N deposition exceeds
nature's tolerance (Dentener et al. 2006; Lamarque et al., 2005; Phoenix et al., 2006).
Here we present a nitrate-focused study that has been organized as a part of the series of
AeroCom phase III experiments (https://wiki.met.no/aerocom/phase3-experiments). The
goals of this activity are to (1) address the diversity of the nitrate simulation by the
AeroCom multi-models and diagnose the driving processes for the diversity, (2) explore
the uncertainty of the model nitrate simulations constrained against various
measurements from ground station networks and aircraft campaigns, and (3) investigate
how the formation of nitrate changes in different models in response to perturbation on
key precursors and factors that determine nitrate formation. We focus on the first two
objectives in this paper. Such a study directs us on how to improve the representation of



nitrate aerosol formation and size distribution in climate chemistry models and reveals
nitrate effects on global air quality and climate.
Building upon the analysis of the multi-model diversity, three additional sensitivity
experiments are designed using the GMI model to further explore the potential sources
for the diversity on physical and chemical process-level. First, we explore the impact of
pH-dependent $NH_3$ wet deposition on atmospheric $NH_3$ and associated nitrogen species.
We then reveal the importance of mineral dust and sea salt in the nitrate formation and
check the resultant nitrate aerosol size distribution that is particularly important in nitrate
forcing estimation.
The paper is organized as follows. Section 2 introduces the experiment setup including
the used emission inventories and the participating Aerocom models. Observations of
surface tracer concentrations and dry and wet depositions over U.S., Europe, and East
Asia, as well as aircraft measurements in the ARCTAS campaigns are described in
section 3. We present AeroCom model inter-comparison and the model evaluation using
aforementioned observations in section 4. Based on the knowledge from previous
sections, we further discuss nitrate formation in response to physiochemical
methodologies in section 5 and summarize our major findings in section 6.
**2. Experiment setup and AeroCom model description**
**2.1 Experiment setup**
The AeroCom III nitrate experiment comprises one baseline and six perturbation
simulations, with the latter designed for assessing the possible future changes of emission
and meteorological fields relevant to nitrate formation. Models are advised to use the
same prescribed emission datasets for gases and aerosols. Emissions from anthropogenic,
aircraft, and ship are obtained from the recently developed HTAP v2 database (Janssens-
Maenhout et al., 2015) that provides high spatial resolution monthly emission. For the
tracers that are included in ozone chemistry but are not provided by HTAP v2 (i.e. some
volatile organic compounds), they should be obtained from CMIP5 RCP85 with a linear
interpolation between 2005 and 2010. Biomass burning emissions are the emissions of
GFED3 (Werf et al., 2010) in 2008 [http://www.globalfiredata.org/data.html]. The $NH_3$
emission from ocean is adopted based on the compilation of GEIA emission inventory
[Bouwman et al., 1997]. Participating modeling groups use their own emissions of
dimenthyl sulfide (DMS), dust, sea salt, and NO from lightning, since they are calculated
based on models' meteorological fields.
A full year simulation for 2008 is required for the nitrate model experiment. There are
several in-situ observation datasets available in 2008 for model evaluation, including the
surface concentration and deposition measurements over the US (CastNet, AMoN,
NDAP/NTN), Europe (EMEP), and Asia (EANET), and the aircraft measurements of
vertical profiles (e.g. ARCTAS-A, ARCTAS-CARB, and ARCTAS-B). All participating
models are required to use the reanalysis or nudged meteorological data for 2008 and
allow several months spin up for the baseline simulation.



**2.2 AeroCom models**
Nine models participate in the AeroCom III nitrate experiment. Their general nitrate-
related physiochemical mechanisms are summarized in Table 1. Further detailed
information on their thermodynamic equilibrium model (TEQM) is given in Table 2.
The models participating in this study are divided into two groups. Group one (CHASER,
EMAC, INCA, GISS-MATRIX, and GISS-OMA) run chemical fields together with
meteorological fields, while group two (EMEP, GMI, OsloCTM2, and OsloCTM3)
simulate chemical fields using archived meteorological fields. Most models in this study
have a horizontal resolution around 2-3 degrees except EMEP with 0.5 degree.
Vertically, most models cover both the troposphere and the stratosphere with a peak
altitude up to 0.01 hPa except EMEP that extends vertically up to 100 hPa into the
troposphere only.
All models use full gas phase $O_3$-$NO_x$-$HO_x$ chemistry to produce $HNO_3$ and consider the
feedback of nitrate aerosol formation on $HNO_3$ calculation. However, due to the
complexity of chemical mechanisms for organic nitrate compounds and different
recommendations for reaction rates, $HNO_3$ fields produced by the models differ greatly.
This difference propagates into the subsequent gas-aerosol reactions for nitrate formation.
These models are very different in their approaches on gas-aerosol reactions in nitrate
formation. All models consider reactions between $NH_3$ and $HNO_3$. However, models
differ dramatically in whether to include heterogeneous reactions on dust and sea salt
(Table 1). Some account for both, some for only dust or sea salt, and some do not account
for any of them at all. The methods used by the models in accounting for $NH_3$ and
dust/sea salt contributions are also different.
All participating models adopt TEQM to deal with aqueous and solid phase reactions and
gas-aerosol partitioning (Tables 1 and 2). This is based on the assumption that volatile
species in the gas and aerosol phases are generally in chemical equilibrium. However, the
assumption is not always warranted in some cases, as we will discuss in section 5.2. Even
with the TEQM approach, nitrate calculation could differ due to treatments of
equilibrium constants or chemical potentials, solute activity coefficients, water activity,
and relative humidity of deliquescence (RHD). The parameterizations adopted by the
models to deal with multicomponent activity coefficient, binary activity coefficient, and
water activity are given in table 2. GISS-OMA, Oslo-CTM2 and Oslo-CTM3 are special
in that they assume aerosols to be metastable so that the model does not take into account
formation of solids in this study. All other models do consider the effect of the hysteresis
of particle phase transitions. All models also assume that the overall particles are large
enough to neglect the Kelvin effect.
The participating models call the TEQMs in different ways to account for aerosol size
effect. All the TEQMs (ISORROPIA-I, ISORROPIA-II, MARS, RPMIRES, INCA, and
EQSAM3) assume particles to be internally mixed, i.e. all particles of the same size have
the same composition. However, some parent models (CHASER, EMEP, GMI, INCA,
GISS-MATRIX and GISS-OMA) call their TEQMs only once for fine mode aerosol



particles, while the others (EMAC, OsloCTM2 and OsloCTM3) call their TEQMs from
different aerosol size bins. For example, Oslo-CTM2 and Oslo-CTM3 consider a bi-
modal aerosol size-spectrum with two major aerosol modes, fine and coarse, and
calculate gas-aerosol equilibrium partitioning with EQSAM3 first for fine mode and then
for coarse mode. Additionally, to account for kinetic limitations, EMAC calculates the
phase partitioning in two stages. In the first stage, the amount of the gas-phase species
that is able to kinetically condense onto the aerosol phase within the model time step is
calculated, while in the second stage, the TEQM redistributes the mass between the two
phases assuming instant equilibrium (Pringle et al., 2010).
The TEQMs also differ in the chemical components considered. Specifically, the TEQMs
in CHASE, EMEP, GISS-MATRIX, GISS-OMA, GMI and INCA include only species
of sulfate, nitrate, ammonium and their gas, liquid, and solid components. The models
Oslo-CTM2 and Oslo-CTM3 add NaCl and HCl, while the model EMAC further expands
the species by including dust-related crustal material such as $Ca^{2+}$, $K^+$, and $Mg^{2+}$.
These TEQMs differ in their computational approaches as well. Computational efficiency
is a prime consideration for a TEQM that is designed for incorporation into a global air
quality and climate study. To speed up the calculation, TEQMs typically divide the
system into sub-domains based on RH and concentrations of ammonium, sodium, crustal
cations, and sulfate. Corresponding approximation could be adopted for each sub-domain
with the minimum numbers of equilibriums and unknown components. As listed in table
2, the numbers of sub-domains are 4, 5, 4, 2, 3, and 3 for the TEQM ISORRPIA-I,
ISORROPIA-II, MARS, RPMIRES, INCA, and EQSAM3, respectively.
The ways to account for the contribution of dust and sea salt to nitrate formation are also
different. Some models (EMAC, Oslo-CTM3, and Oslo-CTM2) include dust and/or sea
salt components in their TEQM models directly, while some models (EMEP, GISS-
OMA, GMI, and INCA) use an approach of first order loss rate outside their TEQMs to
account for the heterogeneous reactions of $HNO_3$ on the surface of dust and sea salt. For
the latter approach, the gamma rates and their RH dependence adopted by the models
differ as well.
Dry and wet deposition of $NH_3$, ammonium nitrate, and ammonium sulfate are treated
similarly to other gas and aerosol tracers in the models. It is worth pointing out that there
is a different consideration for Henry's law constant of $NH_3$ used by the models. Some
models modify it based on the pH value of cloud water while others do not. We will
discuss the impact of these two treatments on nitrate simulation in section 5.1.
We introduce only the major characteristics of thermodynamic equilibrium models since
this study aims for the evaluation and explanation of overall nitrate diversity among the
GCM/CTM models from all potential aspects. The detailed discussion of the models
chemical mechanism of gas phase reactions and the aerosol optical properties adopted by
the models is also beyond this work. Readers could refer to the references listed in Tables
1 and 2 for any further details.



## 3. Observations

We use surface measurements from ground station networks and aircraft campaigns to evaluate modeled surface concentrations, dry and wet depositions, and vertical distributions of nitrate and related species (Table 3).

### 3.1 Surface measurements of concentrations and deposition rates

Ambient concentrations of sulfur and nitrogen species throughout the US and Canada have been measured by the ground station network CASTNET (Clean Air Status and Trends Network) (Figure 1). The measurements use a 3-stage filter pack with a controlled flow rate. The measurements of CASTNET do not include $NH_3$. AMoN (Ammonia Monitoring Network), measuring concentrations of ambient $NH_3$, has been deployed at CASTNET sites starting from October 2007 using passive samplers. The corresponding tracers' surface concentration measurements over Europe have been conducted by EMEP (The European Monitoring and Evaluation Programme). The measured sites of all these networks are located in rural areas or sensitive ecosystems, representing a larger region by avoiding influences and contamination from local sources. Surface concentrations over East Asia are inferred from the measurement of dry deposition by EANET (Acid Deposition Monitoring Network in East Asia). This network provides acid deposition from a regional monitoring network including 13 countries in East Asia using standardized monitoring methods and analytical techniques.

CASTNET also provides dry deposition of sulfate and nitrogen species. Direct measurements of dry deposition fluxes (D) are expensive so D is calculated as the measured pollutant concentration (C) multiplied by the modeled dry deposition velocity ($V_d$). $V_d$ is either estimated by the Multi-Layer Model fed with measured hourly meteorological data or derived from historical average $V_d$ for sites with discontinued meteorological parameters.

Direct measurements of wet deposition fluxes of sulfate, nitrate, and other ions have also been performed by NADP/NTN (the National Atmospheric Deposition Program / National Trends Network) across the contiguous US, Canada, Alaska, and the US Virgin Islands and EANET over East Asia. Sites are predominantly located away from urban areas and point sources of pollution. Each site has a precipitation chemistry collector and gauge. Both networks can measure wet deposition for a continuous period (weekly for NADP/NTN and daily for EANET), or every precipitation event if using an automated collector (wet-only sampling).

Data is quality assured for all measurements. Measurements over North America use automated screening techniques, semi-annual calibration results, site operator comments, and manual data review. Quality assurance of EMEP is carried out on both the national level and by the Chemical Co-ordinating Centre (CCC). The quality of EMEP measurements is not equal at the national level (Schaap et al., 2002; 2004). Sites in North, Western and Central Europe were generally well equipped and performing, while sites in the rest of Europe suffered from inadequate sampling and calibrating methods due to political and/or economical reasons. The quality of ammonia measurement is relatively low since some laboratories experienced contamination problems (Williams et al., 1992).



Although EANET adopts standardized monitoring methods and analytical techniques,
quality assurance is carried out on the national level.
**3.2 Aircraft measurements of vertical profiles**
Aircraft campaign measurements during the 2008 Arctic Research of the Composition of
the Troposphere from Aircraft and Satellites (ARCTAS) are used to evaluate tracer
vertical distribution simulated by the models (Bian et al., 2013; Jacob et al., 2010). Three
phases of the campaign, ranging from Northern Hemisphere mid-latitude industrial
region (ARCTAS-CARB, June 2008) to high latitude Arctic regions influenced by long-
rang pollution transport (ARCTAS-A, April 2008) and by local boreal biomass burning
(ARCTAS-B, July 2008), provide well encompassing environment observations. All
flights were conducted by the NASA DC-8 aircraft and the flight tracks of these three
phases are presented in Figure 2. An onboard HR-ToF-AMS instrument (Cubison et al.,
2011; DeCarlo et al, 2006) measured fine mode aerosol concentrations (PM1) along the
fight track including $NO_3^-$, $NH_4^+$, and $SO_4^{2-}$ at STP conditions (1013mb and 273.15K) at a
sampling time interval of ~12 seconds. Accuracy estimate of 2-standard deviations, likely
conservative, is 34% for inorganics, dominated by the uncertainty in particle collection
efficiency due to particle bouncing (Huffman et al., 2005).
**4. Model intercomparison and evaluation**
**4.1 AeroCom model inter-comparisons of global distributions and budgets**
**4.1.1 $NH_3$ and $NH_4^+$**
Six models use HTAP2 anthropogenic emissions, two (GISS-MATRIX and GISS-OMA)
use CMIP5 emissions, and one (INCA) uses ECLIPSE emissions. Table 4b shows that
eight models have the annual $NH_3$ emission values within 5% of the value from the
AeroCom experiment recommended emission inventories, but INCA is 11% higher. The
similar emission distributions ensure that the examined inter-model diversities are truly
caused by the differences in physicochemical processes among the models. The
normalized root-mean-square deviation (NRMSD) of $NH_3$ global burden among models
is 1.17 and 0.33 with and without EMAC included. This drastic change in global burden
NRMSD by EMAC is caused by its special treatment of wet deposition. In fact, the
removal of trace gases and aerosol particles by clouds and precipitation in EMAC is not
calculated based on empirically determined, fixed scavenging coefficients, but rather by
solving a system of coupled ordinary differential equations, explicitly describing the
processes involved (Tost et al., 2006). This method resolves feedback mechanisms
between the multi-phase chemistry and transport processes involved. The liquid phase
reaction set used converts all the scavenged NH3 (or HNO3) into $NH_4^+$ (or $NO_3^-$) in the
liquid phase so that at the end everything that is deposited is the total $NH_4^+$ and NH3.
Atmospheric $NH_4^+$ is produced entirely by $NH_3$ chemical transformation. The models
simulate $NH_4^+$ much closer in chemical production (difference less than a factor of 2) than
in lifetime (difference up to a factor of 5.2), indicating removing rates are a key factor in
controlling the global burden of $NH_4^+$. For example, CHASER has a much longer lifetime
of $NH_4^+$ (i.e. 9.8 days versus 4.3 days in average), which indicates a slow deposition





removal of $NH_4^+$ from the atmosphere. Consequently, CHASER simulates a much higher
atmospheric $NH_4^+$ burden than other models.
**4.1.2 $HNO_3$ and $NO_3^-$**
$HNO_3$, an important nitrate precursor, differs by up to a factor of 9 in its global
tropospheric burden among the models (Table 4c). All models simulated $HNO_3$ based on
a full gas phase $O_3$-$HO_x$-$NO_x$ chemistry and coupled it with aerosol chemistry. This
$HNO_3$ diversity will naturally be propagated into the $NO_3^-$ simulation. However, further
discussion of the detailed consideration of full gas-aerosol chemistry for $HNO_3$ diversity
among the models is beyond the scope of this study.
The resultant aerosol product (i.e., $NO_3^-$) does not entirely follow its precursor (i.e.,
$HNO_3$) in terms of global burden: EMEP has very low $HNO_3$ but high $NO_3^-$, two GISS
models (MATRIX and OMA) simulate high $HNO_3$ but low $NO_3^-$, while OsloCTM3 has
an average $HNO_3$ but more than triple high $NO_3^-$ than average (Tables 4a and 4c).
Furthermore, the difference in $NO_3^-$ global burden (up to a factor of 13) is larger than that
of $HNO_3$. Differences in chemical mechanisms of $NO_3^-$ production could be a potential
explanation along with the difference in $HNO_3$ precursor. Unfortunately, only GMI and
INCA provide a detailed $NO_3^-$ chemistry budget analysis. Nevertheless, we can infer that
the total chemical production of $NO_3^-$ must be very low (~ 10Tg) in the two GISS models
while very high (> 100 Tg) in OsloCTM2 and OsloCTM3 based on the reported total
$NO_3^-$ loss. Combining this information with the $HNO_3$ global tropospheric burden (Table
4c), we can further infer that the chemical conversion from $HNO_3$ to $NO_3^-$ must be lowest
in the two GISS models while highest in the two Oslo models. Several factors could
influence this conversion, such as the availability of alkaline species of mineral dust and
sea-salt particles and the physicochemical mechanism of nitrate formation on dust and
sea-salt, availability of $NH_3$ after combining with $SO_4^{2-}$, and the atmospheric
meteorological fields of temperature and relative humidity. More discussions are given in
sections 5.2 and 5.3.
Atmospheric lifetime of $NO_3^-$ differs up to a factor of 4, from about 2 days in GMI and
OsloCTM2 to larger than 7 days in GISS-OMA and GISS-MATRIX. The slower removal
processes in the two GISS models compensate the low chemical production and help to
maintain their $NO_3^-$ atmospheric burden (Figure 3 and Table 4a).
**4.2 Model-observation comparisons**
**4.2.1 Comparisons of surface concentrations over North America, Europe, and East**
**Asia**
Understanding diversity among model simulations and potential physiochemical
processes behind the difference is important but not sufficient. The information has to be
combined with the knowledge of model performance obtained directly from comparisons,
particularly down to processes level, against various measurements to gain a direction of
any improvement. Figures 4a-c show a model-observation comparison for surface
mass/volume mixing ratios of $NO_3^-$, $NH_4^+$, $NH_3$, $HNO_3$, and $SO_4^{2-}$ over North America
(CastNET), Europe (EMEP), and East Asia (EANET). Each point represents a monthly





mean concentration at one observational site. Generally, the agreement between model
and observation is better for aerosol components than for gas tracers (i.e. the precursor
species $NH_3$ and $HNO_3$) over all three regions. All models underestimate $NH_3$ surface
volume mixing ratio with a ratio of model to observation down to 0.14, while most
models overestimate surface $HNO_3$ volume mixing ratio with a ratio up to 3.9 over North
America. The worse performances of $NH_3$ against observations may be also associated to
their relatively lower measurement accuracy, i.e. easier to be contaminated during
measurement (Williams et al., 1992). Among aerosol simulations, model performance is
very similar for $NH_4^+$ and $SO_4^{2-}$, while slightly worse for $NO_3^-$ that is dispersed further
away from the 1:1 line, particularly at low $NO_3^-$ values. The $NO_3^-$ simulation over East
Asia is worst with the average normalized root mean square to be 1.3 and 1.8 higher than
that over North America and Europe, respectively.
**4.2.2 Comparisons of vertical profiles with aircraft measurements during the**
**ARCTAS field campaign**
Evaluation of model performance presented in 4.2.1 for the surface concentrations in the
source regions is highly dependent on the accuracy of the emission inventory. On the
other hand, evaluation using aircraft measurements, particularly over remote regions,
provides further examination of models' physicochemical evolution during transport.
Here we use data from three phases of the ARCTAS aircraft campaign (section 3), and
the results are shown in Figure 5. All model results of $NO_3^-$, $NH_4^+$, and $SO_4^{2-}$ are sampled
along flight track and averaged regionally within 1km vertically for each campaign phase
before comparing with the corresponding aircraft measurements. Note that only EMAC,
EMEP and GMI report daily 3D global tracer concentrations, while the others report
monthly only. Note also that only EMEP and GMI adopt daily biomass burning emission
while the others use monthly emission. To verify the representativeness of monthly mean
concentration in capturing the main features exhibited in model-observation comparisons,
daily and monthly concentrations of the three models are used in the same spatial
sampling to compare with the measurements (see the green lines for daily and red for
monthly in the figure). The comparison keeps its main features as shown when using both
daily and monthly model data.
During ARCTAS-A, which was conducted in April 2008 and was based in Fairbanks,
Alaska, none of the models captures the long-range transport of aerosols primarily from
Asia, which enter Polar Regions at altitudes between 2-7 km (Fig. 3 in Bian et al., 2013).
Except CHASER and EMAC, all models also report a significant underestimation of
$NH_4^+$ and $SO_4^{2-}$ in boundary layer. A previous assessment of pollution transport to the
Arctic indicated that aerosol wet removal plays an important role in the uncertainty of
Arctic aerosols (Shindell et al., 2008). Another potential reason is that some large fire
activities in Siberia during April 2008 (Jacob et al., 2010) may be missed in the GFED3
emission inventory. The underestimation of $SO_4^{2-}$ may help bring up $NO_3^-$ production,
particularly at high altitudes. During ARCTAS-CARB, which was conducted in June
2008 based in Palmdale, California, agreement between model and measurements is
much improved. Almost all models show a rapid vertical decease from surface to free
troposphere, which is consistent with the measurements of $SO_4^{2-}$ and $NH_4^+$, but not $NO_3^-$.
The observation shows a maximum of $NO_3^-$ at about 1.5 km, which is not represented by



any of the models. During ARCTAS-B, which was conducted in July 2008 and was based
in Cold Lake, Canada, when there were frequent local wild fires, model performances are
mixed. In general, most models underestimate concentrations of $NO_3^-$, $NH_4^+$ and
$SO_4^{2-}$ below 4 km. CHASER model is special in that it overestimates $SO_4^{2-}$ significantly.
This may be contributed to high (near surface) to comparable (free troposphere) model
simulation of $NH_4^+$ but an underestimation of $NO_3^-$. Different from other models, the
INCA model shows an enhancement of pollutants in the upper troposphere with
concentrations much higher (more than 5 times) than observations. This behavior may be
derived from a much vigorous vertical uplifting to the upper troposphere as revealed from
Fig. 3a-3b combined with a low $NH_3$ Henry's law constant used by INCA, see discussion
in section 5.2.
Note that all measurements and model data we discussed above are for fine mode
aerosols. Total $NO_3^-$ (orange line using monthly model output) is also shown in the figure
to reveal whether a changing of partitioning of fine and coarse mode $NO_3^-$ could improve
the model-observation comparison. It seems that the new version of OsloCTM3 may put
too much of $NO_3^-$ in coarse mode.
**4.3 Model-observation comparison for dry and wet deposition**
**4.3.1 Dry deposition**
The budget analyses in section 4.1 concluded that dry and/or wet depositions are most
likely the main processes driving the diversity in the model simulations. Thus, further
evaluation of deposition processes is needed to identify any potential problematic model.
The dry depositions of $NO_3^-$, $NH_4^+$, $HNO_3$, and $SO_4^{2-}$ simulated by the models are
compared against CASTNET measurements over North America (Figure 6). Generally,
the overestimation of surface $HNO_3$ concentrations (Figure 3a) results in the higher dry
depositions of $HNO_3$, but this is not the case for $NO_3^-$. Meanwhile, most of the models
give a better dry deposition simulation for aerosol $SO_4^{2-}$ and $NH_4^+$ than for aerosol $NO_3^-$,
except CHASER. Specifically, GISS-OMA and GISS-MATRIX have wide spread dry
$NO_3^-$ deposition at any given measurement value. In other words, the two models
underestimate $NO_3^-$ dry deposition significantly at many observational stations, which
does not occur in the other models. This low dry deposition simulation may occur outside
North America as well because the global dry depositions of the two models are lower
than others (Table 4a). OsloCTM2 overestimates $NO_3^-$ dry deposition significantly, which
is probably linked to its larger coarse fraction of the nitrate aerosol (see discussion in
section 5.3). OsloCTM3 improved its dry deposition scheme although the model still
overestimates the dry deposition. We will discuss the OsloCTM2 $NO_3^-$ simulation over
North America by combining the model's wet deposition in the next section. $NH_4^+$ dry
deposition is low in GMI but very high in CHASER. This performance is also extended
globally as summarized in Table 4b.
**4.3.2 Wet deposition**
The wet deposition simulations from the nine models are compared with surface
measurement over North America (Figure 7a) and East Asia (Figure 7b) for oxidized





$NO_3^-$ (i.e. total $NO_3^-$ and $HNO_3$), total $NH_4^+$ and $NH_3$ ($tNH_4^+$), and $SO_4^{2-}$. All models tend
to underestimate the wet deposition of $tNH_4^+$ and $SO_4^{2-}$ over the two regions. Models
EMAC, GMI, OsloCTM2 and OsloCTM3 have relatively high wet removal for oxidized
$NO_3^-$, while EMEP removes much less than others over North America. All models' wet
deposition of oxidized $NO_3^-$ is biased low over East Asia. As we discussed above,
OsloCTM2 and OsloCTM3 have very high dry $NO_3^-$ depositions (Figure 6) compared
with CASTNET observations. The overall high dry and wet $NO_3^-$ depositions along with
high atmospheric concentrations (Figure 4a) indicate that the chemical formation of
$NO_3^-$ in the two models must be also high. This performance might be also true on global
scale since the inferred chemical productions of $NO_3^-$ in the two models are the highest
(Table 4a). CHASER has the lowest $tNH_4^+$ wet deposition. This may result in a very high
$NH_4^+$ dry deposition (Figure 6) and concentration (Figures 4a-c, 5) compared with
observations and other models. Overall, wet deposition seems to be the dominant process
in determining the diversity in $NH_3$ and $NH_4^+$ lifetime (Table 4b).
Note that we use the traditional approach of comparing models' grid box mean values
with observations, which does not take into account the impact of the models' horizontal
resolutions in their representation of observations (Schutgens et al., 2016). Since majority
models (except EMEP) have horizontal resolutions around 2-3 degrees, the models grid
box means tend to smooth out extreme (i.e. very low or high) observations. Reflecting on
the scattering plots of model (y-axis) and observation (x-axis) is that the slopes of fitting
lines are generally less than 1 (Figures 4a-d, 6, 7a-b).
**5. Discussion of major uncertainties in nitrate formation**
Large uncertainties of nitrate studies result from the complexity of the simulations which
must consider a comprehensive $NO_x$-NMHC-$O_3$-$NH_3$ chemistry and a thermodynamic
equilibrium model (TEQM) to partition semi-volatile ammonium nitrate between the gas
and aerosol phases. Nitrate aerosol concentrations depend on temperature, relative
humidity (RH), and concentrations of $HNO_3$, $NH_3$, $NH_4^+$, $SO_4^{2-}$, $Cl^-$, $Na^+$, $Ca^{2+}$, $K^+$,
$Mg^{2+}$, organic acids, among others. A further complicating factor is that the equilibrium
for the coarse mode is somewhat questionable (Feng and Penner, 2007). In addition, wet
removal of $NH_3$ is very sensitive to the correction of pH in cloud water. We will discuss
some of these uncertainties below.
**5.1 pH-dependent $NH_3$ wet deposition**
Gas tracer $NH_3$, a precursor of ammonium aerosol, experiences atmospheric wet
deposition and its deposition rate is typically calculated using Henry's Law. Henry's law
constant (H) of gases in water is usually given at 298 K (indicated by Θ in superscript)
and can be adjusted by temperature (T).

$$H(T) = H^\Theta * \exp\left(-\frac{\Delta H_{sol}}{R}\left(\frac{1}{T} - \frac{1}{T^\Theta}\right)\right) \qquad (1)$$

Here $\Delta H_{sol}$ is the enthalpy of dissolution and R is the gas constant.
For some acidic/basic gases, including $NH_3$, Henry's law constant is also a function of
pH in cloud water (a.k.a effective Henry's law constant $H^{\Theta*}$). As explained in the
Appendix, the $H^{\Theta*}$ is inferred from $H^\Theta$ with a correction of pH (pH = $-\log_{10}[H^+]$) as

$$H^{\Theta *} = H^{\Theta} \frac{K_{al}[H^+]}{K_w} \qquad (5)$$

Here, $K_{al} \approx 1.8 \times 10^{-5}$ and $K_w = 1.0 \times 10^{-14}$ at 298 K in pure water (see Appendix). However,
not every model accounts for pH adjustment (i.e. the reaction of equation 2 in Appendix)
for $NH_3$ dissolution. More accurately, the EMAC model implicitly calculates the
effective Henry's law constant by solving a set of partial differential equations, which
includes not only the gas-liquid phase equilibria, but also the reactions in the liquid phase
(i.e. dissociation or acid-base equilibria, Redox reactions and photolysis reactions in the
liquid phase, see Tost et al.(2006)). Therefore, the gas-liquid phase equilibrium is
explicitly calculated based on the chemical mechanism used in the liquid phase. As listed
in Table 5, the rest of the models are generally divided into two groups based on their
effective Henry's law constant: (1) INCA, GISS-OMA and GISS-MATRIX has $H^{\Theta *} \le$
100 (L-theta without pH correction) and (2) CHASER, GMI, OsloCTM2 and Oslo-
CTM3 has $H^{\Theta *} > 10^{+5}$ (H-theta with pH correction). The $NH_3$'s $H^{\Theta *}$ adopted by the
models varies dramatically, up to an order of 6 in magnitude among all the models and a
factor of 10 just for the models in H-theta group (Table 5). The latter is corresponding to
a correction for pH ranging from 4.5 (Oslo-CTM2) to 5.5 (CHASER).
To examine how sensitive of $NH_3$, $NH_4^+$ and $NO_3^-$ simulations in response to the
magnitude of $NH_3$'s $H^{\Theta *}$, we performed a sensitivity experiment, named TWET, in the
GMI model in which there was no pH adjustment for $NH_3$ Henry's law constant (i.e.
$H^{\Theta *}$=61 instead of 1.05e+6, see table 6). The resultant annual budgets of dry/wet
deposition, chemistry production and loss, and atmospheric loading of $NH_3$,
$NH_4^+$ and $NO_3^-$ are summarized in Table 7, the tracers' vertical zonal mean distributions
are shown in Figure 8, and the comparisons with the ARCTAS measurements for
$NH_4^+$ and $NO_3^-$ are shown in Figure 9. For convenient comparison, the GMI baseline
results are given in the table and figures as well. There is a dramatic decrease (from 17.5
to 1.1 Tg) in $NH_3$ wet deposition when using pure water $NH_3$ Henry's law constant.
Consequently, $NH_3$ will remain in the atmosphere (i.e. ~ 8 times more atmospheric $NH_3$)
to produce ~1.6 times more $NH_4^+$ chemically. This, in turn, greatly increases atmospheric
$NO_3^-$ to 0.97 Tg from 0.26 Tg reported in baseline simulation. A large portion of the
increased $NH_3$, $NH_4^+$ and $NO_3^-$ resides in the upper troposphere and close to the
tropopause region, while the changes of the tracers in the lower troposphere are relatively
small, as shown in Figure 8. These accumulations at high altitudes are far above (i.e. ~ 50
times for $NH_4^+$ and $NO_3^-$) the ARCTAS observed tracer amounts as shown in Figure 9.
The TWET experiment might be an explanation of $NH_4^+$ and $NO_3^-$ accumulations near the
tropopause region (Figure 3a-b) in the INCA model whose $NH_3$ Henry's law constant $H^{\Theta}$
is 74 without pH correction (i.e. a L-theta model, table 5). However, it is puzzling that the
$NH_3$ simulations by GISS-MATRIX and GISS-OMA, those are the models with L-theta,
are closer to the simulations of the models with H-theta, i.e. no $NH_4^+$ and $NO_3^-$
accumulation near the tropopause and comparable removal of $NH_4^+$ (Figure 3a-b and
Table 4b).
**5.2 Contribution of dust and sea salt on nitrate formation**
In the presence of acidic accumulation-mode sulfuric acid containing aerosols, $HNO_3$,
$NO_3$ radicals, and $N_2O_5$ will deposit on larger alkaline mineral or salt particles (Dentener



et al., 1996; Gard et al., 1998; Hauglustaine 2014; Karydis et al., 2016; Murphy and
Thomson 1997; Paulot et al., 2016). Considerable evidence shows that the majority of
atmospheric nitrate is formed via reactions associated with dust and sea salt (Allen et al.
2015; Itahashi et al., 2016; Karydis et al., 2016). Coarse mode nitrate overwhelmingly
dominates over remote oceanic regions (Itahashi et al., 2016). Over wide land regions,
nitrate also quite often exists in the form of supermicron $NO_3^-$ balanced by the presence
of mineral cations arising from transport of crustal dust and sea spray aerosol (Allen et
al.,2015; Lefer and Talbot; 2001).
Investigation of nitrate interactions with mineral dust and sea salt depends on the
simulation approach adopted in a model. The traditional equilibrium approach to partition
semi-volatile $HNO_3$ between the gas and aerosol phases is no longer possible since the
time to reach equilibrium on coarse mode particles (several hours to days) is typically
much longer than the chemical time step used in a global model (less than 1 hour) (John
et al., 1989; Myhre et al., 2006). Meng and Seinfeld (1996) found that on longer time
scales, when $NH_3/HNO_3$ started to condense on larger aerosols, their gas phase
concentrations decreased so that some of the condensed matter can be driven back to the
gas phase from the small semi-volatile aerosols. A fix to a non-equilibrium state would
be to implement a kinetic formulation for the particles that have a long equilibrium time
scale (Feng and Penner, 2007; Karydis et al., 2010). However, implementing explicit
kinetics in a global model would be computationally expensive and, hence, is not feasible
for long-term climate simulations. Several approximations, therefore, have been
developed to compromise accuracy and efficiency.
Four such approximations are adopted by the nine models participating in this study: 1)
using equilibrium calculations for fine mode particles only while neglecting nitrate
formation on coarse mode particles (CHASER and GISS-MATRIX); 2) combining
equilibrium calculation for a solution of $SO_4^{2-}$-$NO_3^-$-$NH_4^+$-$H_2O$ and heterogeneous
reaction calculation for nitrogen uptake on dust and sea-salt using a first-order loss rate
(EMEP, GMI, GISS-OMA and INCA); 3) running equilibrium model including $NH_3$,
dust and sea salt repeatedly for aerosol sizes from fine mode to coarse mode (Oslo-CTM2
and Oslo-CTM3); and 4) using only the fraction of the gas that can kinetically condense
within the time step of the model in the equilibrium calculations for each aerosol size
mode (EMAC).
Nitrate is formed primarily on dust and sea salt by GMI (88%) and INCA (82%) (see
Table 4a). INCA further separates the formation as 45% on dust and 37% on sea-salt. The
above-mentioned approach 1 is problematic due to absence of coarse mode nitrate, an
important portion of nitrate, which results in relatively low nitrate burdens for CHASER
and GISS-MATRIX. Unfortunately, the other models are missing a detailed nitrate
chemistry budget report. A potential impact of dust and sea-salt on nitrate formation,
nevertheless, can be inferred from the approach adopted by a model. For example,
OsloCTM2 and OsloCTM3 adopt approach 3. Although the model allows fine mode
particles to reach equilibrium first, the subsequent equilibrium calculation for coarse
mode particles may still produce coarse mode nitrate too quickly, see discussion of the
ratio of coarse model nitrate in the next subsection. To avoid such overestimations on the




production of coarse mode nitrate, EMAC allows only a fraction of $HNO_3$ to partition in
the aerosol phase by assuming diffusion limited condensation (Pringle et al., 2010).
To further understand the role of homogeneous and heterogeneous chemical reaction
processes in nitrate formation, we conducted two more sensitivity experiments,
TnoCNH3 and TnoCHET, with the GMI model (Table 6). Experiment TnoCNH3 turned
off chemical conversion of $NH_3$ to $NH_4^+$ in the GMI thermodynamic equilibrium model,
while experiment TnoCHET excluded the nitrate formation via heterogeneous reaction of
gas $HNO_3$ on the particles of dust and sea salt. The budget report, vertical zonal mean
distribution and model-observation comparison of $NH_3$, $NH_4^+$ and $NO_3^-$ are given in Table
7 and Figures 8-9, respectively. It is not surprising that experiment TnoCNH3 gives a
higher atmospheric $NH_3$ burden (0.32 Tg) compared with baseline (0.11 Tg) with little
$NH_4^+$ left (from its initial field). The interesting thing is that the formed $NO_3^-$ has only
slightly decreased compared with baseline (from 0.26 to 0.20 Tg), confirming the
importance of $NO_3^-$ formation via dust and sea salt. For experiment TnoCHET, the
simulations of $NH_3$ and $NH_4^+$ stay the same but the formed $NO_3^-$ is decreased dramatically
(from 0.26 to 0.10), indicating that $NO_3^-$ formation via $NH_3$ chemistry alone in the GMI
model is relatively small. The chemical production of $NO_3^-$ is about 6 times larger in
TnoCNH3 (via dust and sea salt) than in TnoCHET (via $NH_3$). However, the $NO_3^-$
produced via $NH_3$ chemistry (TnoCHET) is non-negligible over remote regions impacted
by long-range transport, as shown in the analysis of April Alaska observations in Figure
664    9.
**5.3 Nitrate size distribution**
Unlike sulfate aerosol, a noticeable fraction of nitrate aerosol is in the coarse mode.
Having an accurate aerosol size distribution is critical in climate forcing estimations,
since large size particles have a relatively small optical cross section at a given aerosol
mass loading and the nitrate material coating on dust particles has almost no direct impact
on the dust optics, although the greatly impact dust lifetime (Bauer et al., 2007). Given
that the deposition velocity of a coarse particle is greater than that of a fine particle, an
accurate size distribution is also necessary to estimate deposition of particulate nitrates
(Yeatman et al., 2001; Sadanaga et al., 2008). This estimation is particularly important
over oceans where coarse mode nitrate dominates (Itahashi et al., 2016) and nitrogen
supply is often in deficit (Hansell and Follows, 2008).
As we have discussed in section 5.2, nitrate size distribution varies with the approaches
adopted for nitrate formation on coarse mode aerosols (i.e. dust and sea salt). Figure 10
gives the burdens of nitrate in fine mode and coarse mode portions and the ratio between
coarse mode and total (f_c) for the eight discussed models. The ratio is ranging from 0
(CHASER and GISS-OMA), ~50% (EMAC, GMI and INCA), ~80% (EMEP and
OsloCTM2), and 97% (OsloCTM3). The two OsloCTMs give the highest f_c partially
because they run TEQM model for coarse model particles.
A wide range of f_c, from 0 to > 90%, has been reported previously by model
simulations (Adams et al., 2001; Bauer et al., 2007; Jacobson 2001), while the range is
narrowed down to 40-60% for the model studies using the approach that solves dynamic





mass transfer equation for coarse mode particles (Feng and Penner, 2007; Xu and Penner,
690   2012).
It is worth pointing out that aerosol microphysics modify aerosol size as well. For
example, a process like coagulation would also allow $NO_3^-$ to mix with other particles and
enter coarse mode aerosol. New particle formation/nucleation would add $NH_3/NH_4^+/NO_3^-$
into the ultra fine mode. Except EMAC and GISS-MATRIX, majority models involved in
this study are bulk aerosol models that do not account for aerosol microphysics.
It is challenging to verify the nitrate size distribution globally due to the limited
measurements on time and space. Measurements over regional and station sites indicated
that the ratio of f_c could be very high and vary seasonally over oceanic sites. For
example, annual mean f_c during 2002-2004 from the Fukue supersite observatory is
about 72% with a seasonal variation of 60–80% in winter and of around 80% in summer
(Itahashi et al., 2016).
However, the ratio could be varied dramatically over land or the areas affected by land
pollution. For example, observations of fine and coarse particulate nitrate at several rural
locations in the United States indicated that nitrate was predominantly in submicron
ammonium nitrate particles during the Bondville and San Gorgonio (April) campaigns, in
coarse mode nitrate particles at Grand Canyon (May) and Great Smoky Mountains
(July/August), and both fine and coarse mode nitrate during the studies at Brigantine and
San Gorgonio (July) (Lee et al., 2008). Allen et al. (2015) examined aerosol composition
data collected during the summer 2013 SOAS and concluded that inorganic nitrate in the
southeastern United States likely exists in the form of supermicron $NO_3^-$, balanced by the
presence of mineral cations arising from the transport of crustal dust and sea spray
aerosol. The measurements over Harvard Forest, a rural site in central Massachusetts,
supported that the majority of nitrate mass was associated with water-soluble
supermicron soil-derived $Ca^{2+}$ in an acidic environment (Lefer and Talbot, 2001).
Measurements of coarse-mode aerosol nitrate and ammonium at two polluted coastal
sites, Weybourne, England and Mace Head, Ireland, during polluted flow when the air
had passed over strong source regions of the UK and northern Europe, showed 40–60%
of the nitrate was found in particles with diameter >1 μm, but under clean marine
conditions almost 100% conversion was seen (Yeatman et al., 2001).
**6. Conclusions**
We present the AeroCom phase III nitrate study by assessing aerosol simulations of
nitrate and ammonium and their precursors with nine global models. Five of the models
couple the chemical calculation online with meteorological simulation, and four use
archived meteorological fields driving chemistry. To focus on chemical-physical
processes behind the diversity of nitrate simulation, all participating models are
encouraged to use HTAP2 emission inventory. The simulated aerosols of nitrate and
ammonium and their precursors are compared among the models and evaluated against
various measurements including surface concentrations and dry/wet depositions from
surface measurements, and vertical distributions from aircraft measurements.



All models capture the main features of the distribution of nitrate and ammonium: large
surface and column amounts over China, South Asia, Europe, and U.S. These regions are
typically densely populated with large $NH_3$ and $NO_x$ emissions. Many models also show
enhanced nitrate and ammonium over the Middle East and continents over the Southern
Hemisphere. The former undergoes huge dust pollution and the latter experiences fires
that emit both $NH_3$ and $NO_x$.

The diversity of nitrate and ammonium simulations among the models is large: the ratio
of the maximum to minimum quantities among the nine models is 13.4 and 4.4 for model
simulated global mass burdens of nitrate and ammonium, respectively, and 3.9 and 5.2
for the corresponding lifetimes. These values are also larger than those of sulfate: 4.0 for
global burden and 3.0 for lifetime.

The agreement between models and observations is better for aerosol components than
for gas tracers. All models underestimate $NH_3$ surface mass concentrations but most
models overestimate surface $HNO_3$ concentrations over North America and East Asia.
Performance of $NH_3$ is the worst: this could partially be associated to its relatively lower
measurement accuracy, i.e. a loss of ammonia possibly on the filters designed to collect
$NH_3$ (Williams et al., 1992). Among aerosol simulations, model performance based on
evaluation of surface mixing ratio and dry/wet depositions is very similar for $NH_4^+$ and
$SO_4^{2-}$, while slightly worse for $NO_3^-$. Models severely underestimate the aerosol
concentrations with only a few exceptions when compared with aircraft measurements
and this problem is worse over regions impacted by long-range transport than those
closer to sources.

There are many intrinsic reasons for a larger diversity in nitrate simulations among
models. Nitrate is involved in much more complicated chemistry: the chemical
mechanism needs to handle a multiphase multicomponent solution system. The system
sometimes cannot even be solved using the thermodynamic equilibrium approach when
coarse mode dust and sea salt particles present. A reasonable nitrate simulation also
depends on good simulations of various precursors, such as $NH_3$, $HNO_3$, dust and sea
salt, although models account for impact of dust and sea salt very differently. Even an
accurate simulation of $SO_4^{2-}$ is a prerequisite because $SO_4^{2-}$ surpasses $NO_3^-$ at reacting
with $NH_4^+$.

The models' intercomparison and model-observation comparison revealed at least two
critical issues in nitrate simulation that demand further exploration: $NH_3$ wet deposition
and relative contribution to $NO_3^-$ formation via $NH_3$ and dust/sea salt. The nine
participating models adopt very different effective Henry's law constants for $NH_3$, with
one group having a value equal or less than 100 (in pure water) and the other larger than
1.e+05 (with pH correction). Sensitivity studies using the GMI model indicated that
without pH correction, $NH_3$ wet deposition decreases massively (from 17.5 to 1.1 Tg),
which prolongs atmospheric $NH_3$ lifetime (from 0.67 to 5.2 days) and enhances its
atmospheric burden (from 0.11 to 0.85 Tg), and thus the atmospheric burden of $NH_4^+$
(from 0.17 to 0.48 Tg) and $NO_3^-$ (from 0.26 to 0.97 Tg) as well. These enhanced tracers





tend to accumulate in the upper troposphere and close to the tropopause, and are too high
when compared with aircraft measurements.
All the models use thermodynamic equilibrium to solve the chemical process of
$NH_3/NH_4^+$ to $NO_3^-$ formation in fine model aerosols. However, the models adopt very
different ways in accounting for the contribution of these reactions on the surface of dust
and sea salt particles: some account for both dust and sea salt, some account for only dust
or only sea salt, and two models even do not account for any heterogeneous reactions.
The methodologies that take dust and sea salt into account are also very different, i.e.
together with $NH_4^+$ using thermodynamic equilibrium model or simply adopting a first
order loss rate on dust and sea salt surfaces. The chemical budget reported by GMI and
INCA indicates that the majority (>80%) of global $NO_3^-$ formation is via reaction on dust
and sea salt. Two sensitivity experiments using the GMI model by tagging the $NO_3^-$
formation from either $NH_3/NH_4^+$ chemistry or heterogeneous reactions on dust and sea
salt confirm the critical importance of the latter process, and indicate that the former
process is relatively important in remote regions. The importance of $NO_3^-$ formation on
dust and sea salt lies also in its determination on nitrate particle size distribution, so that
has an implication in air quality and climate studies as well.
Our work presents a first effort to assess nitrate simulation from chemical and physical
processes. A companion study is proposed by AeroCom III nitrate activity to investigate
how sensitive is nitrate formation in response to the possible future changes of emission
and meteorological fields. These perturbation fields include increasing $NH_3$ emission,
decreasing NOx, SOx and dust emissions, and increasing atmospheric temperature and
relative humidity. Based on the findings of this work, modelers should pay particular
attention to incorporating dust and sea salt and treating $NH_3$ wet deposition to improve
nitrate simulation. Further evaluation using satellite measurements, such as $NH_3$ products
from IASI and TES, is desired and will be conducted. Such evaluation requires global 3-
dimensional high frequency model data. Potential future study also includes estimation of
nitrate forcing for climate change.

**Appendix**
For some acidic/basic gases, including $NH_3$, Henry's law constant is also a function of
pH in water (a.k.a effective Henry's law constant). This is because not only does the
aqueous chemistry reaction $NH_3 + H_2O$ (equation 1) reach equilibrium within a chemical
time step but its product $NH_3 \cdot H_2O$ (equation 2) does as well.

$$NH_3 + H_2O \Leftrightarrow NH_3 \cdot H_2O \qquad (1)$$
$$NH_3 \cdot H_2O \Leftrightarrow NH_4^+ + OH^- \qquad (2)$$

Here, $NH_4^+$ is the ammonium ion and OH⁻ is the hydroxide ion. The total dissolved
ammonia $[NH_3^T]$ is given by

$$\begin{aligned}
[NH_3^T] &= [NH_3 \cdot H_2O] + [NH_4^+] \\
&= p_{NH3} H^\Theta \left( 1 + \frac{K_{al}[H^+]}{K_w} \right) \\
&\approx p_{NH3} \left( H^\Theta \frac{K_{al}[H^+]}{K_w} \right) \qquad (3)
\end{aligned}$$



Here, $p_{NH3}$ is the partial pressure of NH₃, $K_{al} = [NH_4^+][OH^-] / [NH_3 \bullet H_2O] \approx 1.8 \times 10^{-5}$, and
$K_w = 1.0 \times 10^{-14}$ at 298 K in pure water. So the effective Henry's law constant $H^{\Theta*}$ is
inferred from $H^{\Theta}$ with a correction of pH (pH = $-\log_{10}[H^+]$) as

$$H^{\Theta*} = H^{\Theta} \frac{K_{al}[H^+]}{K_w} \qquad (4)$$

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




**Table 1. Nitrate chemical mechanism and physical properties of AeroCom**
**models**

| Model | CHEM-EQM | HNO3 chem mechanism | CHEM DUST | CHEM SEASALT | How do CHEMDUSS | Bins for nitrate | Model Name & resolution | References |
|---|---|---|---|---|---|---|---|---|
| CHASER | ISORROPIA-I | CHASER (Sudo et al., 2002) | No | No | --- | Fine mode | MIROC, GCM, 2.8°x2.8°x64 | Watanabe et al., 2011 |
| EMAC | ISORROPIA-II (Stable state[a]) | MESSy2 (Jöckel et al., 2010) | Yes | Yes | ISORROPIA-II | 4 bins: Nucleation, Aitken, Accumulation, Coarse | ECHAM5, GCM, 2.8°x2.8°x31 | Karydis et al., 2016 |
| EMEP | MARS | EMEP EmChem09 (Simpson et al., 2012) | Yes | Yes | First order loss | Fine and coarse | ECMWF-IFS, CTM, 0.5x0.5°x20 | Simpson et al., 2012 |
| GMI | RPMARES (Stable state) | GMI (Straham et al., 2007) | Yes | Yes | first order loss | 3 bins: (D<0.1, 0.1 – 2.5, > 2.5 um) | MERRA2, CTM, 2.5°x2°x72 | Bian et al., 2009 |
| INCA | INCA (Stable state) | INCA tropospheric chemistry (Hauglustaine et al., 2004) | Yes | Yes | first order loss | 2 bins : (D< 1µm and 1 - 10µm) | LMD-v4, GCM, 1.9°x3.75°x39 | Hauglustaine et al., 2014 |
| GISS MATRIX | ISORROPIA-II (Stable state) | MATRIX Bauer (2008) and tropospheric chemistry (Shindell et al., 2003) | No | No | NO | Distributed over all mixing states e.g. size distributions. | NASA GISS-E2, GCM, 2°x2.5°x40 | Schmidt et al 2014 |
| GISS OMA | EQSAM_v03d (Metastable[b]) | OMA (Bauer 2007) and tropospheric chemistry (Shindell et al., 2003) | Yes | No | Bauer and Koch, 2005 | Fine mode | NASA GISS-E2, GCM, 2°x2.5°x40 | Schmidt et al 2014 |
| Oslo CTM2 | EQSAM_v03d (Metastable) | Oslo CTM2 (Berntsen and Isaksen, 1997) | No | Yes | EQSAM_v03d | 2 bins: fine and coarse mode | ECMWF, CTM, 2.8°x2.8°x60 | Myhre et al., 2006 |
| Oslo CTM3 | EQSAM_v03d (Metastable) | Oslo CTM2 (Berntsen and Isaksen, 1997) | No | Yes | EQSAM_v03d | 2 bins: fine and coarse mode | ECMWF, CTM, 2.25°x2.25°x60 | Myhre et al., 2006 |

[a]Stable state: where salts precipitate once the aqueous phase becomes saturated
[b]Metastable: where the aerosol is composed only of a supersaturated aqueous phase
**Table 2. Characteristics of thermodynamic equilibrium models**

| | ISORROPIA-I | ISORROPIA-II | MARS | RPMARES | INCA | EQSAM_v03d |
|---|---|---|---|---|---|---|
| Species | Sulfate, nitrate, ammonium, sodium, chloride | Sulfate, nitrate, ammonium, sodium, chloride, crustal species | Sulfate, nitrate, ammonium | Sulfate, nitrate, ammonium | Sulfate, nitrate, ammonium | Sulfate, nitrate, ammonium, sodium, chloride |
| # of components | 23 | 34 | 16 | 11 | 9 | 18 |
| # of reactions | 15 | 27 | 7 | 6 | 4 | 25 |
| Multicomponent activity coefficient | Bromley | Bromley | Bromley | Bromley | Seinfeld and Pandis | Metzger |
| Binary activity coefficient | Kusik and Meissner | Kusik and Meissner | Pitzer | Pitzer | Seinfeld and Pandis | Metzger |
| Water activity | ZSR[a] | ZSR | ZSR | ZSR | | ZSR |
| Kelvin effect | No | No | No | No | No | No |
| Quantities that determine subdomains | $[Na^+]$, $[NH_4^+]$, $[SO_4^{2-}]$ | $[Ca^{2+}]$, $[K^+]$, $[Mg^{2+}]$, $[Na^+]$, $[NH_4^+]$, $[SO_4^{2-}]$ | RH, $[NH_4^+]$, $[SO_4^{2-}]$ | $[NH_4^+]$, $[SO_4^{2-}]$ | $[NH_4^+]$, $[SO_4^{2-}]$ | $[NH_4^+]$, $[SO_4^{2-}]$ |
| # of subdomains | 4 | 5 | 4 | 2 | 3 | 3 |




ᵃZSR: *Zdanovskii-Stokes-Robinson*

**Table3. Summary of the observational data used in this study**

| SURFACE NETWORK | QUANTITY | COVER AREA | # of sites in 2008 | SAMPLE FREQUENCE | SOURCE |
|---|---|---|---|---|---|
| CASTNET | Concentration of $HNO_3$, $NO_3^-$, $NH_4^+$, $SO_4^{2-}$ Dry deposition of them | North America | 83 | weekly | www.epa.gov/castnet/ clearsession.do |
| AMoN | Concentration of $NH_3$ | U.S. | 19 | 2-weekly | http://nadp.isws.illinois.edu/ |
| NADP/NTN | Wet deposition of $HNO_3+NO_3^-$, $NH_4^+$, $SO_4^{2-}$ | U.S. | 253 | weekly | nadp.isws.illinois.edu |
| EMEP | Concentration of $HNO_3$, $NH_3$, $NO_3^-$, $NH_4^+$, $SO_4^{2-}$ | Europe | 35 | daily | http://www.nilu.no/projects/ccc/index.html |
| EANET | Concentration of $HNO_3$, $NH_3$, $NO_3^-$, $NH_4^+$, $SO_4^{2-}$ | East Asia | 56 | Daily to 2-weekly | http://www.eanet.asia/eanet/brief.html |
|  | Wet deposition of $HNO_3+NO_3^-$, $NH_4^+$, $SO_4^{2-}$ |  |  | 24 hours or precipitation event |  |
| AIRCRAFT CAMPAIGNS | QUANTITY | COVER AREA | # of Flights | CAMPAIGN PERIOD | SOURCE |
| ARCTAS-A | Concentration of $NO_3^-$, $NH_4^+$, $SO_4^{2-}$ | Alaska, U.S. | 11 | March-April | http://www-air.larc.nasa.gov/cgi-bin/arcstat-c |
| ARCTAS-CARB |  | California Bay area U.S. | 6 | June |  |
| ARCTAS-B |  | Central Canada | 7 | July |  |



**Table 4a. $NO_3^-$ global budget for each model**

| Tracer | Model | Burden (Tg) | SConc (µg kg⁻¹) | DDep (Tg a⁻¹) | WDep (Tg a⁻¹) | ChemDUSSᵃ (Tg a⁻¹) | ChemPᵃ (Tg a⁻¹) | Lifetime (days) | AODᵇ |
|---|---|---|---|---|---|---|---|---|---|
| $NO_3^-$ | CHASER | 0.16 | 0.18 | - | - | - | - | - | 0.0076 |
|  | EMAC | 0.67 | 0.47 | 46.3 | - |  | - | - | - |
|  | EMEP | 0.96 | 0.30 | 15.0 | 62.7 | (71.7)ᶜ | 4.5 | 4.5 | 0.0073 |
|  | GISS-MATRIX | 0.22 | 0.06 | 1.3 | 9.6 | (10.9) |  | 7.4 | - |
|  | GISS-OMA | 0.14 | 0.05 | 1.1 | 5.5 | (6.6) |  | 7.8 | 0.0153 |
|  | GMI | 0.26 | 0.22 | 14.9 | 31.5 | 41.9 | 4.8 | 2.1 | 0.0047 |
|  | INCA | 0.79 | 0.17 | 4.5 | 44.6 | 44.1 | 9.8 | 5.9 | 0.0064 |
|  | Oslo-CTM2 | 0.60 | 0.25 | 47.8 | 61.5 | (109.3) |  | 2.0 | 0.0018 |
|  | Oslo-CTM3 | 1.88 | 0.36 | 34.6 | 90.6 | (125.2) |  | 5.5 | - |
|  | Avg | 0.63 | 0.23 | 20.7 | 45.9 | 60.6 |  | 5.0 | 0.0072 |
|  | Med | 0.60 | 0.22 | 15.0 | 44.6 | 46.7 |  | 5.5 | 0.0064 |
|  | Ratioᵈ | 13.4 | 9.4 | 43.5 | 16.5 | 19.0 |  | 3.9 | 8.5 |

ᵃ: ChemDUSS and ChemP refer to $NO_3^-$ chemical production associated with dust/sea
salt and $NH_3/NH_4^+$, respectively
ᵇ: AOD here includes $NH_4^+$ that is associated to $NO_3^-$ for all models expect EMEP
ᶜ: value inside parenthesis is estimated total $NO_3^-$ chemical production based on its
total loss, while budget without parenthesis is reported directly by model.
ᵈ: a ratio between maximum to minimum model simulations




**Table 4b NH$_3$ and NH$_4^+$ global budget for each model**

| Tracer | Model | Emi (Tg a⁻¹) | Burden (Tg) | SConc (µg kg⁻¹) | DDep (Tg a⁻¹) | WDep (Tg a⁻¹) | ChemP/L (Tg a⁻¹) | Lifetime (days) | AOD |
|---|---|---|---|---|---|---|---|---|---|
| NH$_4^+$ | CHASER | | 0.75 | 0.44 | 20.9 | 7.2 | (28.1)[a] | 9.8 | - |
| | EMAC | | 0.19 | 0.12 | 3.6 | 44.5[b] | - | - | - |
| | EMEP | | 0.20 | 0.15 | 4.0 | 26.4 | (30.4) | 2.4 | 0.0059 |
| | GISS-MATRIX | | 0.31 | 0.18 | 4.1 | 27.9 | (32.0) | 3.5 | - |
| | GISS-OMA | | 0.31 | 0.19 | 4.2 | 24.0 | (28.2) | 4.0 | - |
| | GMI | | 0.17 | 0.14 | 1.7 | 30.6 | 32.2 | 1.9 | - |
| | INCA | | 0.39 | 0.08 | 2.4 | 20.4 | 22.9 | 6.3 | - |
| | Oslo-CTM2 | | 0.29 | 0.14 | 5.3 | 32.6 | (37.9) | 2.8 | - |
| | Oslo-CTM3 | | 0.30 | 0.16 | 5.6 | 26.1 | (31.7) | 3.5 | - |
| | Avg | | 0.32 | 0.18 | 5.8 | 24.4[c] | 30.4 | 4.3 | |
| | Med | | 0.30 | 0.15 | 4.1 | 26.3[c] | 31.1 | 3.5 | |
| | Ratio | | 4.4 | 5.5 | 12.3 | 4.5[c] | 1.7 | 5.2 | |
| NH$_3$ | CHASER | 62.8 | 0.13 | 0.46 | 19.8 | 6.8 | (36.2)[a] | 0.76 | |
| | EMAC | 59.3 | 0.85 | 1.39 | 15.5 | - | - | - | |
| | EMEP | 56.9 | 0.09 | 0.46 | 15.4 | 18.2 | (33.6) | 0.98 | |
| | GISS-MATRIX | 63.4[d] | 0.17 | 0.26 | 18.1 | 13.4 | (31.9) | 0.98 | |
| | GISS-OMA | 63.4[d] | 0.17 | 0.25 | 18.4 | 16.7 | (28.3) | 0.98 | |
| | GMI | 60.4 | 0.11 | 0.40 | 12.6 | 17.5 | 30.4 | 0.67 | |
| | INCA | 70.5[d] | 0.12 | 0.39 | 29.3 | 18.6 | 22.4 | 0.62 | |
| | Oslo-CTM2 | 65.9 | 0.08 | 0.27 | 15.8 | 8.1 | (42.0) | 0.44 | |
| | Oslo-CTM3 | 63.3 | 0.05 | 0.51 | 23.7 | 7.7 | (31.9) | 0.29 | |
| | Avg | 62.9 | 0.20 | 0.49 | 18.7 | 13.4 | 32.1 | 0.72 | |
| | Med | 63.3 | 0.12 | 0.40 | 18.1 | 15.1 | 31.9 | 0.72 | |
| | Ratio | 1.2 | 17.0 | 5.6 | 2.3 | 2.7 | 1.9 | 3.4 | |

[a] chemical budgets inside parenthesis are inferred based on the reported emission
and total deposition
[b] EMAC gives total wet deposition of NH$_4^+$ and NH$_3$
[c] Statistic values of NH$_4^+$ wet deposition do not include EMAC
[d] INCA uses ECLIPSE anthropogenic emissions, two GISS models use CMIP5
anthropogenic emission, and all other models use HTAPv2 anthropogenic emissions

**Table 4c. HNO$_3$ global budget for each model**

| Tracer | Model | Burden[a] (Tg) | SConc (µg kg⁻¹) | DDep (Tg a⁻¹) | WDep (Tg a⁻¹) | CheAP (Tg a⁻¹) | CheGP (Tg a⁻¹) | CheAL (Tg a⁻¹) | CheGL (Tg a⁻¹) | Lifetime (days) |
|---|---|---|---|---|---|---|---|---|---|---|
| HNO$_3$ | CHASER | 1.1 | 0.29 | 74.0[a] | 120.9[b] | - | - | - | - | - |
| | EMAC | 3.1 | 0.32 | 56.1 | 136.0[b] | - | - | - | - | - |
| | EMEP | 0.66 | 0.04 | 39.2 | 123.9 | - | - | - | - | - |
| | GISS-MATRIX | 5.7 | 0.12 | 61.7 | 167.5 | - | - | - | - | - |
| | GISS-OMA | 5.3 | 0.10 | 49.8 | 148.2 | - | - | - | - | - |
| | GMI | 1.8 | 0.18 | 39.7 | 128.1 | 128.1 | 413 | 42.6 | 299 | 3.5 |
| | INCA | 1.5 | 0.09 | 47.7 | 77.5 | 21 | 369 | 10.0 | 210 | 5.7 |
| | Oslo-CTM2 | 1.3 | 0.05 | 36.1 | 66.0 | | | | | |
| | Oslo-CTM3 | 2.3 | 0.04 | 36.0 | 49.3 | - | - | - | - | - |
| | Avg | 2.5 | 0.14 | 45.8[b] | 108.7[c] | | | | | |
| | Med | 1.8 | 0.10 | 43.7[b] | 123.9[c] | | | | | |



| | Ratio | 8.6 | 8.0 | 1.6[b] | 3.4[b] | | | | |
|---|---|---|---|---|---|---|---|---|---|

[a]HNO3 burden for the atmosphere up to 100 hPa
[b]for both $HNO_3$ and $NO_3^-$
[c]statistical values do not include CHASER and EMAC that report total dry or wet
deposition of $HNO_3$ and $NO_3^-$
**Table 4d. $SO_4^{2-}$ global budget for each model**

| Tracer | Model | Emi SO2 (Tg a-1) | Emi SO4 (Tg a-1) | Burden (Tg) | SConc (µg kg-1) | DDep (Tg a-1) | WDep (Tg a-1) | Chem GP (Tg a-1) | Chem AqP (Tg a-1) | Lifetime (days) | AOD |
|---|---|---|---|---|---|---|---|---|---|---|---|
| $SO_4^{2-}$ | CHASER | 111 | 0 | 3.3 | 1.44 | 22.1 | 137 | (159) | | 7.6 | 0.0826 |
| | EMAC | 138 | 619[a] | 1.9 | 1.72 | 504[b] | 302 | (187) | | 0.86 | - |
| | EMEP | 109 | 0 | 0.83 | 0.45 | 10.2 | 109 | (119) | | 2.5 | 0.0232 |
| | GISS-MATRIX | 133 | 5.1 | 1.3 | 0.63 | 16.6 | 97 | (109) | | 4.2 | - |
| | GISS-OMA | 133 | 5.1 | 1.1 | 0.53 | 11.8 | 112 | 52.7 | 66.2 | 3.3 | 0.0714 |
| | GMI | 111 | 0 | 1.1 | 0.58 | 7.5 | 205 | 126.5 | 86.1 | 3.6 | 0.0257 |
| | INCA | 116. | 8.0 | 1.8 | 0.34 | 8.4 | 116 | 42.2 | 75.1 | 5.3 | 0.0417 |
| | Oslo-CTM2 | 133 | 4.1 | 2.0 | 0.49 | 17.6 | 184 | (198) | | 3.6 | 0.0366 |
| | Oslo-CTM3 | 133 | 4.1 | 2.7 | 0.55 | 20.2 | 160 | (176) | | 5.5 | |
| | Avg[c] | 122 | | 1.8 | 0.63 | 14.3 | 140 | 151 | | 4.5 | 0.0469 |
| | Med[c] | 133 | | 1.6 | 0.54 | 14.2 | 127 | 139 | | 3.9 | 0.0392 |
| | Ratio[c] | 1.2 | | 4.0 | 4.2 | 2.9 | 2.1 | 2.0 | | 3.0 | 3.6 |

[a] EMAC emission also includes sea spray $SO_4^{2-}$
[b] EMAC dry deposition includes sedimentation and $SO_4^{2-}$ sedimentation is very high
since it has assumed that 7.7% of sea salt is $SO_4^{2-}$
[c] Statistical values do not include EMAC
**Table 5: Effective Henry Law constant used in the models**

| Aerocom Model | $H^{\ominus*}$ (M/atm) | $-\Delta H_{sol}/R$ (K) |
|---|---|---|
| CHASER | 3.0e+5 | 3400 |
| EMAC[a] | – | – |
| EMEP[b] | – | – |
| GIS MATRIX | 1.e+2 | 3415 |
| GISS OMA | 1.e+2 | 3415 |
| GMI | 1.05e6 | 4200 |
| INCA | 7.4e+1 | 3400 |
| Oslo-CTM2 | 3.3e+6 | 0 |
| Oslo-CTM3 | 3.3e+6 | 0 |

[a]EMAC:  See its wet deposition description in section 4.1.1.
[b]EMEP: The model does not use Henry law but applies simple empirical scavenging
ratio, which for $NH_3$ is 1.4e6 for in-cloud and 0.5e6 for below-cloud scavenging. The
scavenging ratio by definition is the ratio the concentration of a certain pollutant in
precipitation divided by the concentration of the pollutant in air.






**Table 6. Baseline and three sensitivity experiments in the GMI model**

| Experiment | Setup | Purpose |
|---|---|---|
| **BASE** | Standard simulation as described in section 2.1 | Baseline simulation |
| **TWET** | Set $NH_3$ effective Henry law constant from 1.05e+6 (pH= 5.0) to 62 (pure water) | Review impact of $NH_3$ wet deposition |
| **TnoNH3** | Turn off $NO_3^-$ production from $NH_3/NH_4^+$ | Identify how large/where the $NO_3^-$ formation from $NH_3/NH_4^+$ |
| **TnoHET** | Turn off $NO_3^-$ production from dust and sea salt | Identify how large/where the $NO_3^-$ formation from dust and sea salt |


**Table 7: $NO_3^-$, $NH_4^+$, $NH_3$ and $HNO_3$ budgets from the base simulation and three**
**sensitivity experiments**

| Tracer | Exps | Burden (Tg) | SConc (µg kg⁻¹) | DDep (Tg a⁻¹) | WDep (Tg a⁻¹) | ChemDUSS (Tg a⁻¹) | ChemP( Tg a⁻¹) | Lifetime (days) |
|---|---|---|---|---|---|---|---|---|
| $NO_3^-$ | BASE | 0.26 | 0.22 | 14.9 | 31.5 | 41.9 | 4.8 | 2.1 |
| | Twet | 0.97 | 0.23 | 14.8 | 43.3 | 41.0 | 18.3 | 6.0 |
| | TnoNH3 | 0.20 | 0.17 | 14.7 | 27.5 | 42.3 | 0 | 1.7 |
| | TnoHET | 0.099 | 0.065 | 0.61 | 6.70 | 0 | 7.1 | 5.0 |


| Tracer | Model | Emi (Tg a⁻¹) | Burden (Tg) | SConc (µg kg⁻¹) | DDep (Tg a⁻¹) | WDep (Tg a⁻¹) | ChemP/L (Tg a⁻¹) | Lifetime (days) |
|---|---|---|---|---|---|---|---|---|
| $NH_4^+$ | BASE | | 0.17 | 0.14 | 1.7 | 30.6 | 32.2 | 1.9 |
| | Twet | | 0.48 | 0.16 | 1.9 | 50.7 | 53.0 | 3.4 |
| | TnoNH3 | | - | - | - | - | - | - |
| | TnoHET | | 0.17 | 0.14 | 1.6 | 30.6 | 32.2 | 1.9 |
| $NH_3$ | BASE | 60.4 | 0.11 | 0.40 | 12.6 | 17.5 | 30.4 | 0.67 |
| | Twet | | 0.85 | 0.81 | 8.70 | 1.1 | 50.1 | 5.2 |
| | TnoNH3 | | 0.32 | 0.58 | 20.9 | 39.3 | 0 | 1.9 |
| | TnoHET | | 0.10 | 0.40 | 12.6 | 17.4 | 30.4 | 1.2 |







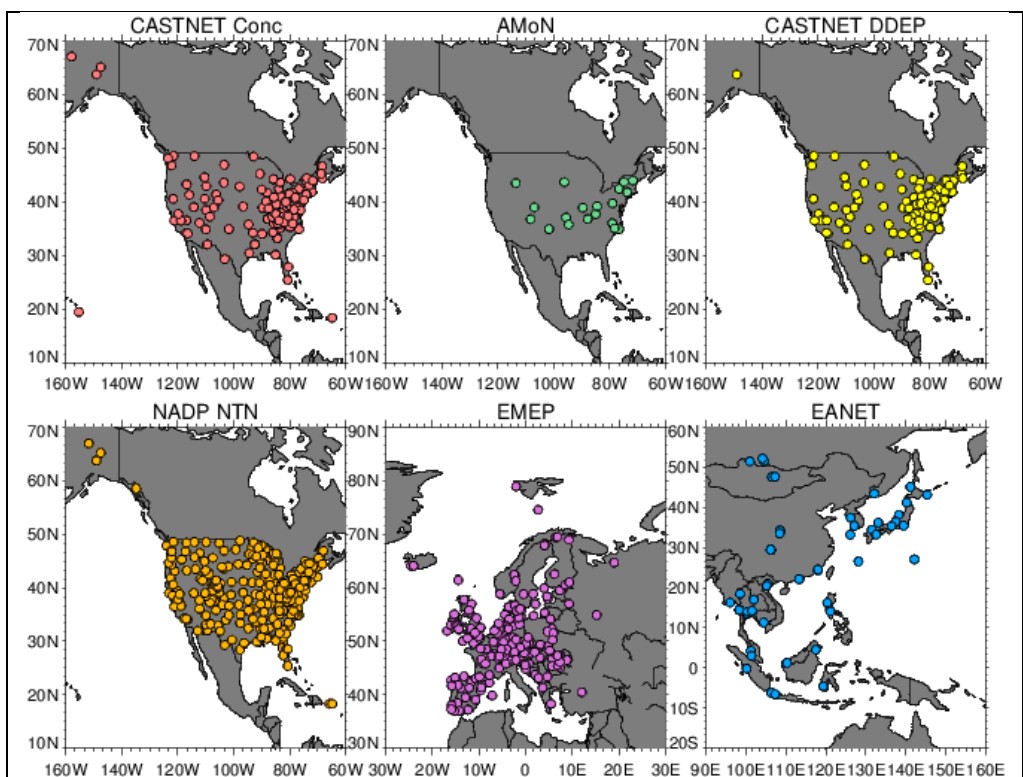

Figure 1. The observational station locations for CASTNET surface concentrations (CASTNET Conc), Ammonia surface monitor network over U.S. (AMON), CASTNET dry deposition (CASTNET DDEP); National Acid Deposition Network for wet deposition over U.S. (NADP NTN), surface concentrations over Europe (EMEP), and surface dry and wet deposition over Asia (EANET).




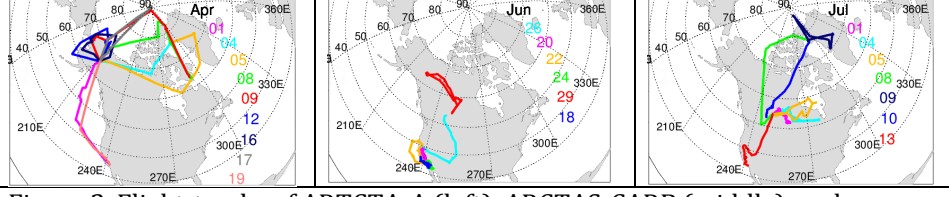

Figure 2. Flight-tracks of ARTCTA-A (left), ARCTAS-CARB (middle), and ARCTAS-B (right). The colors represent observations during different days.


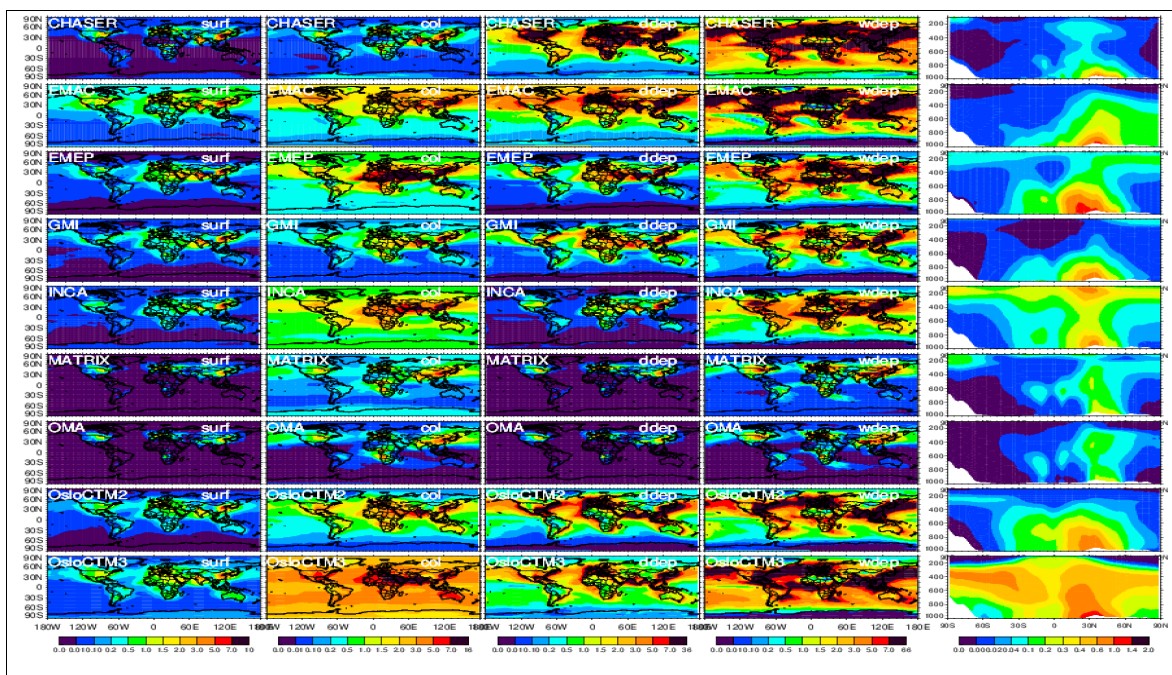

Figure 3a. Multimodel comparison of $NO_3^-$ for surface mass mixing ratio (µg kg$^{-1}$, left), column load (mg m$^{-2}$, second), dry deposition (ng m$^{-2}$ s$^{-1}$, third), wet deposition (ng m$^{-2}$ s$^{-1}$, fourth), and vertical zonal mean (0.5µg kg$^{-1}$, right). Note that the CHASER dry and wet depositions and the EMAC wet deposition in this figure contain both $NO_3^-$ and $HNO_3$, while the rest models $NO_3^-$.




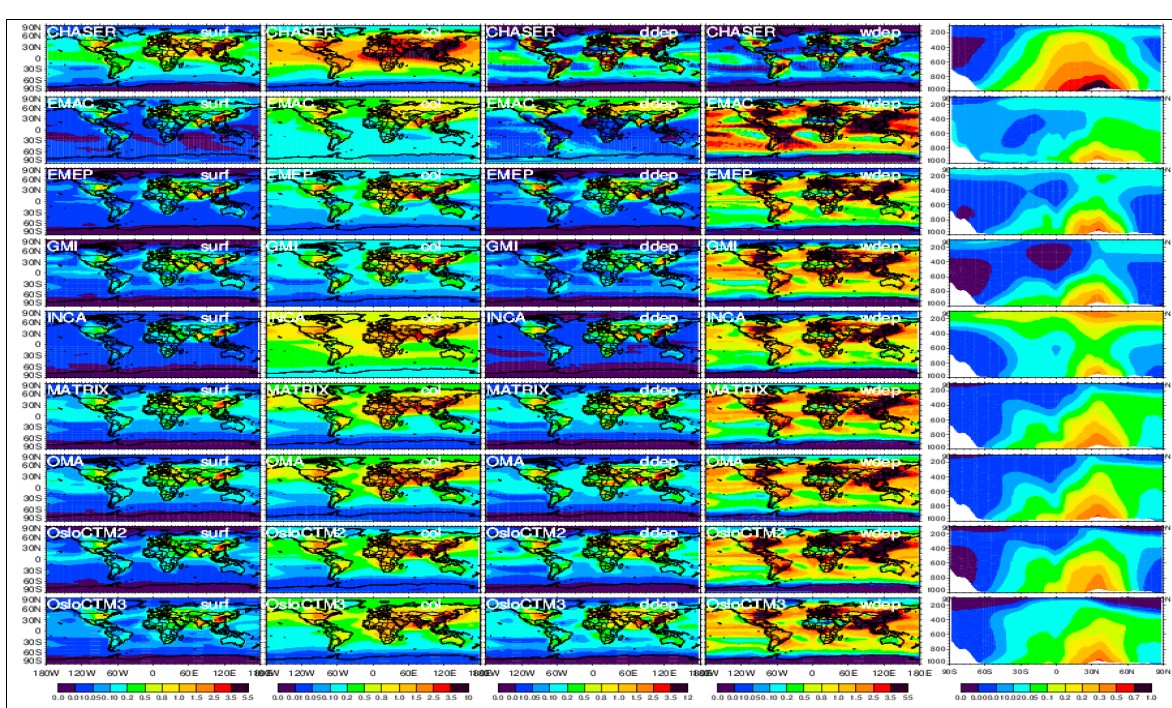

Figure 3b. Same as Figure 3a but for $NH_4^+$ and the unit in vertical distribution is µg kg$^{-1}$. Note that the EMAC wet deposition in this figure contain both $NH_4^+$ and $NH_3$, while the rest models only $NH_4^+$.



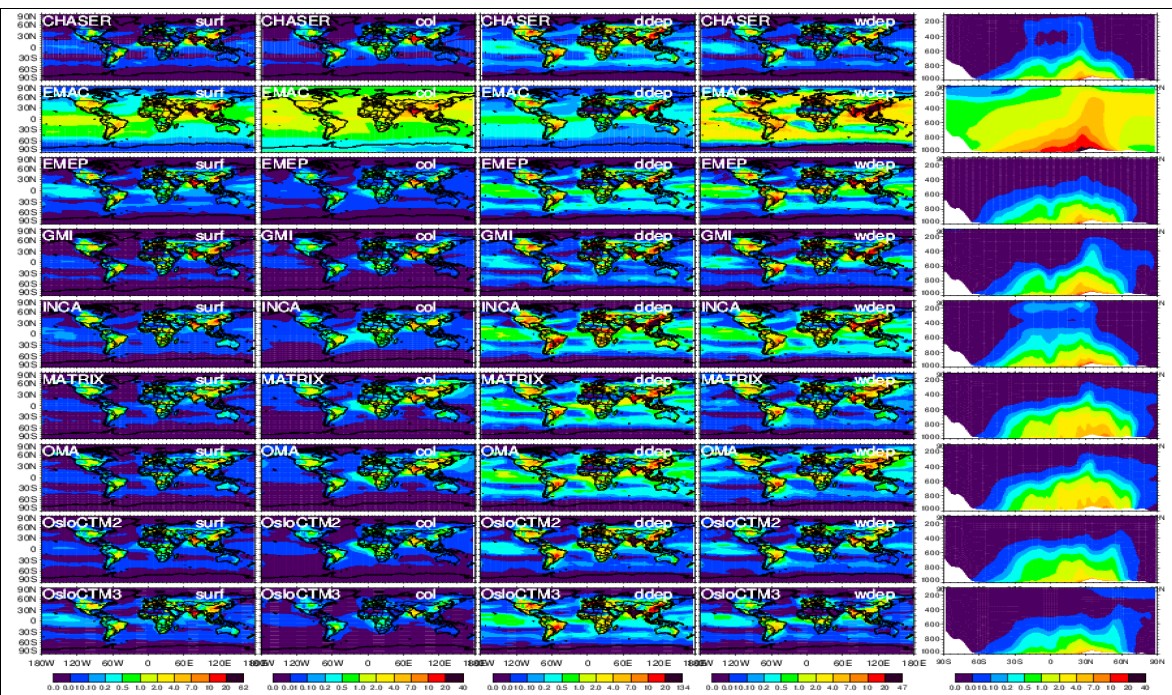

Figure 3c. Same as Figure 3a but for NH$_3$. Units are ppb for surface concentration and 0.1ppb for vertical distribution. Note that the EMAC wet deposition in this figure contain both NH$_3$ and NH$_4^+$, while the rest models only NH$_3$.



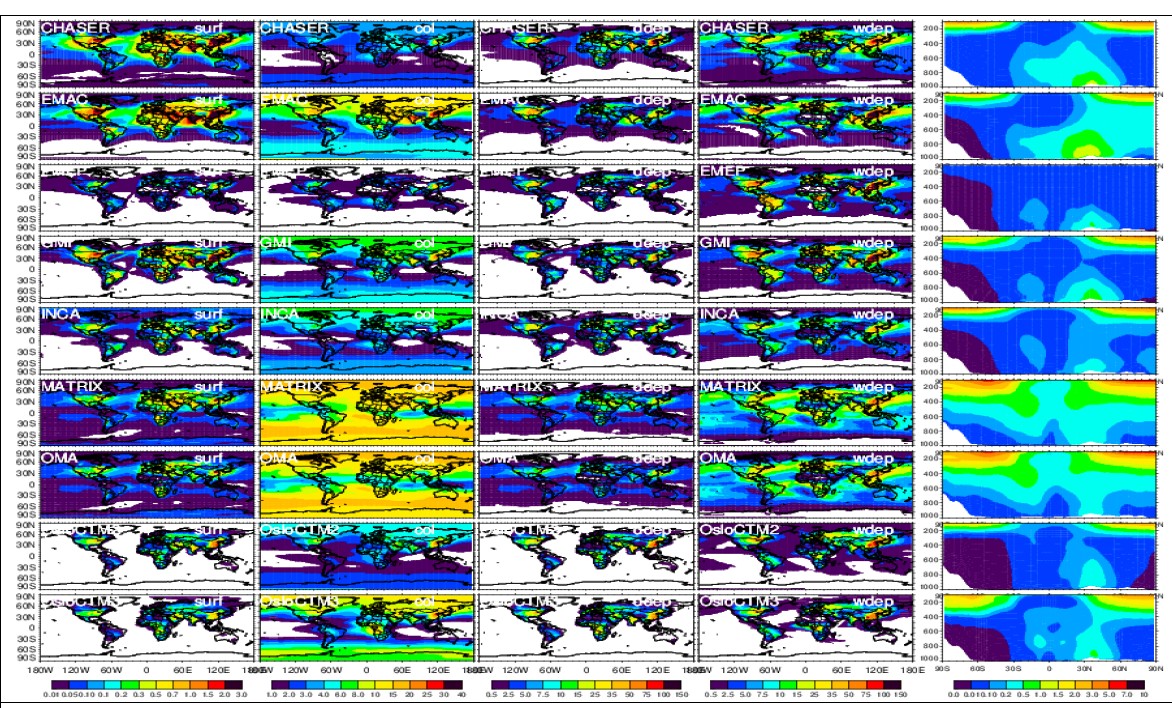

Figure 3d. Same as Figure 3a but for HNO$_3$. Units are 100 ppb for surface concentration, mg m$^{-2}$ for loading, and 2ng m$^{-2}$ s$^{-1}$ for dry and wet depositions. Note that the column total of HNO$_3$ is from surface up to 100 ppb vertically. The CHASER dry and wet depositions and the EMAC wet deposition in this figure contain both HNO$_3$ and NO$_3^-$, while the rest models only HNO$_3$.





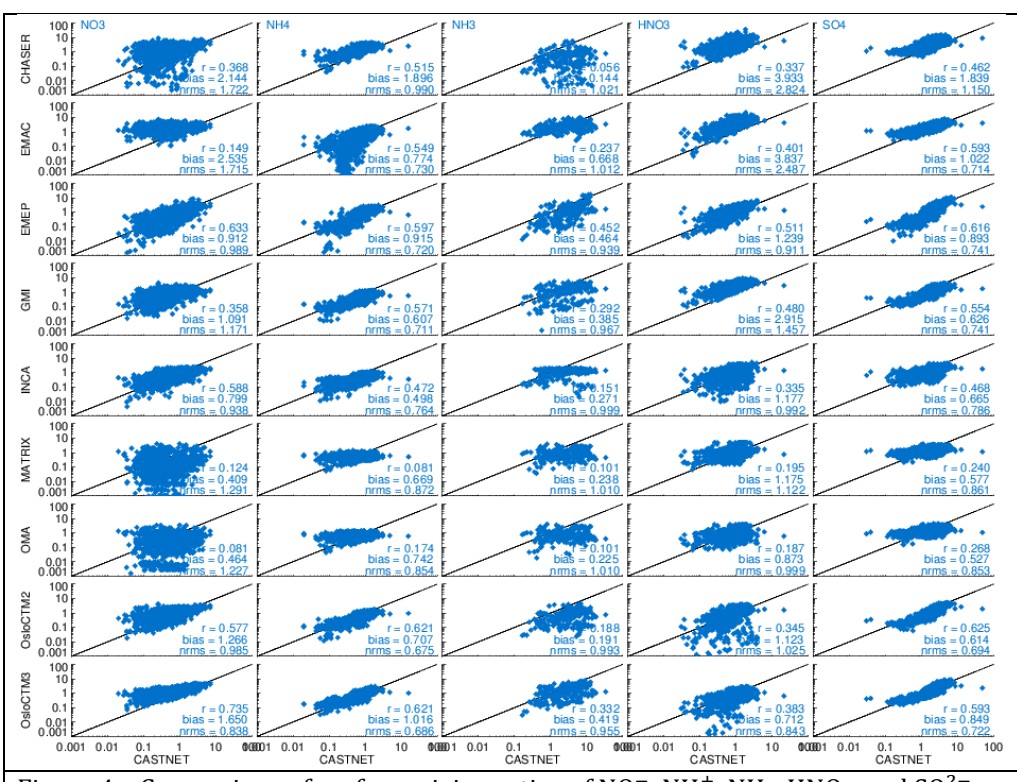

Figure 4a. Comparison of surface mixing ratios of $NO_3^-$, $NH_4^+$, $NH_3$, $HNO_3$, and $SO_4^{2-}$ between models and CASTNET measurement.  Units are µg m$^{-3}$.





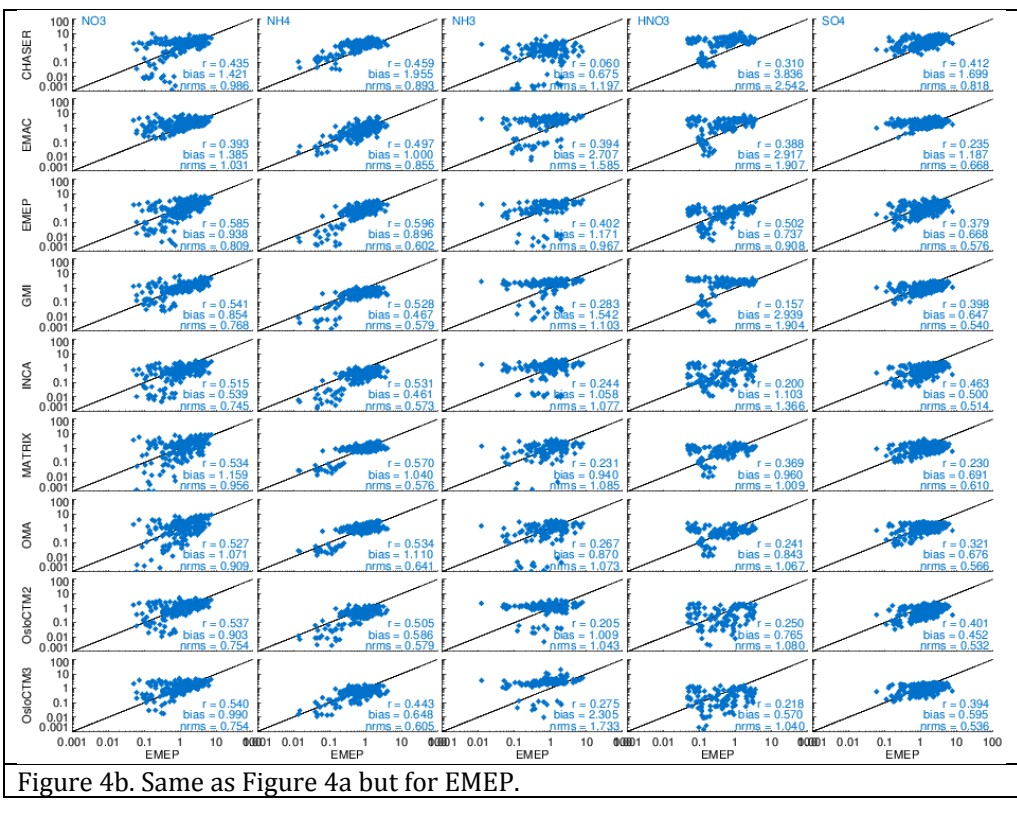

Figure 4b. Same as Figure 4a but for EMEP.



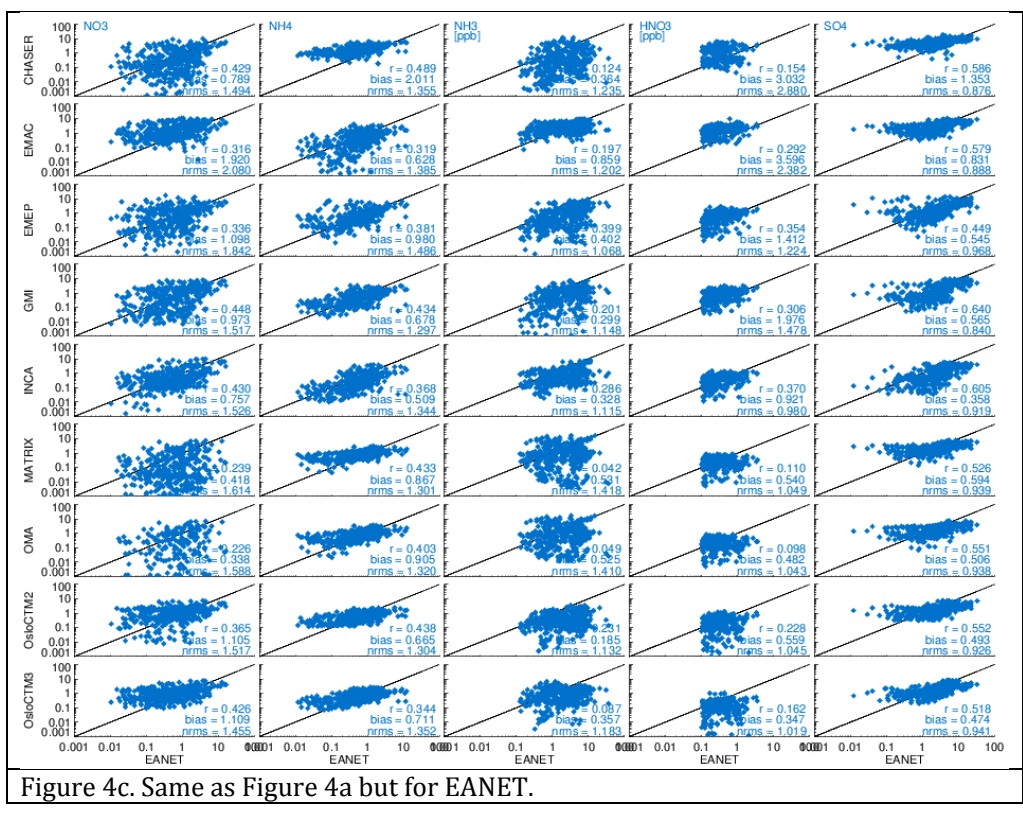

Figure 4c. Same as Figure 4a but for EANET.





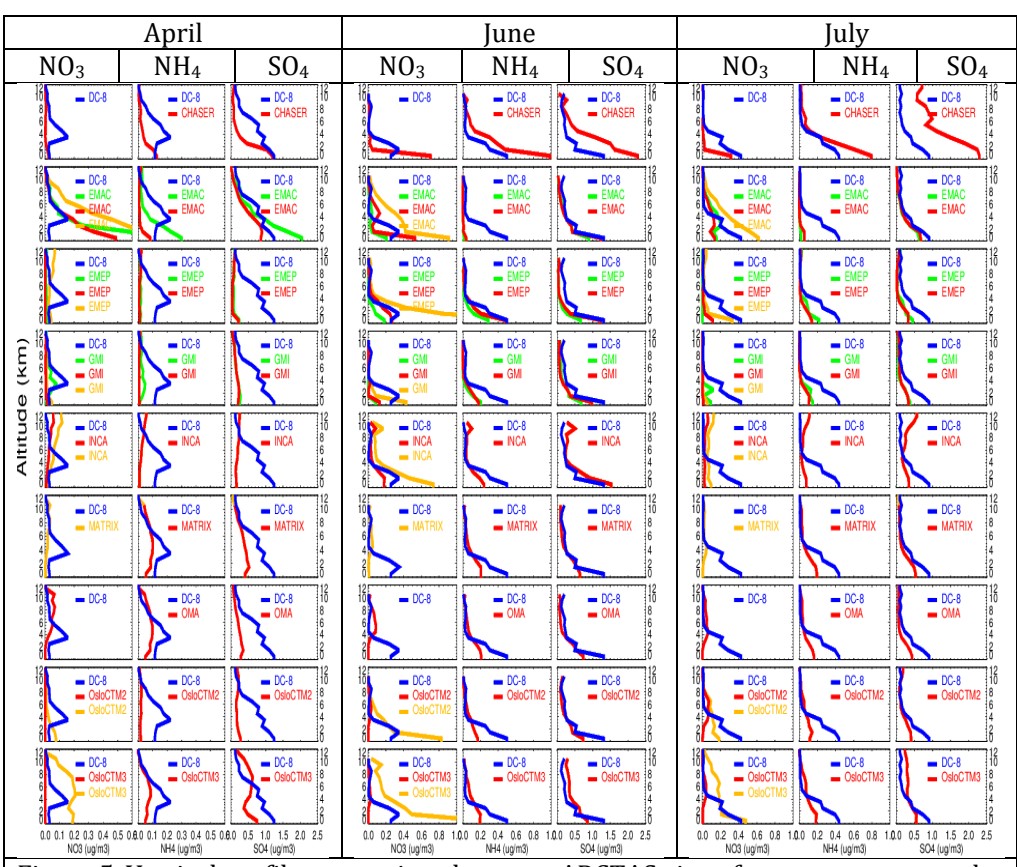

Figure 5. Vertical profile comparison between ARCTAS aircraft measurements and AeroCom model simulations. Note that ARCTAS AMS measurements give fine mode aerosols. Model profiles are shown in green (fine mode aerosol analyzed with daily output), red (fine mode aerosol with monthly output), and orange (total $NO_3^-$ with monthly output). CHASER and OMA have fine mode $NO_3^-$ only. Units are µg m$^{-3}$.





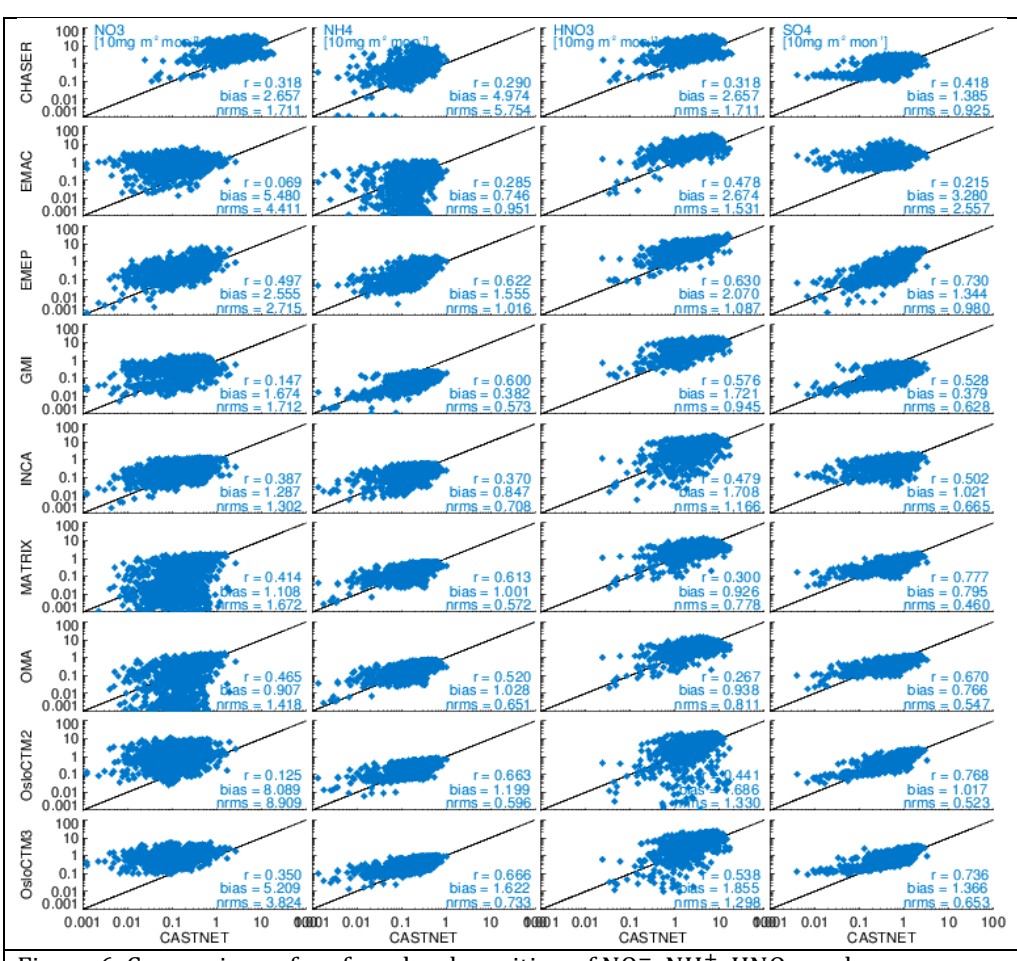

Figure 6. Comparison of surface dry deposition of $NO_3^-$, $NH_4^+$, HNO$_3$, and $SO_4^{2-}$ between models and CASTNET measurements. Units are 10mg m$^{-2}$ mon$^{-1}$.



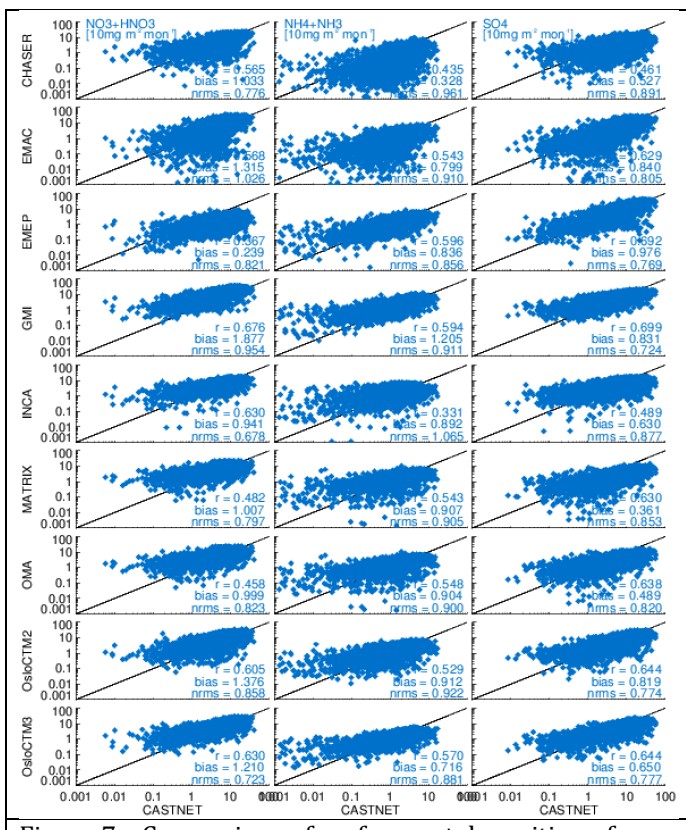

Figure 7a. Comparison of surface wet deposition of $NO_3^- + HNO_3$, $NH_4^+ + NH_3$, and $SO_4^{2-}$ between models and NDAP/NTN measurements. Units are 10mg m$^{-2}$ mon$^{-1}$.





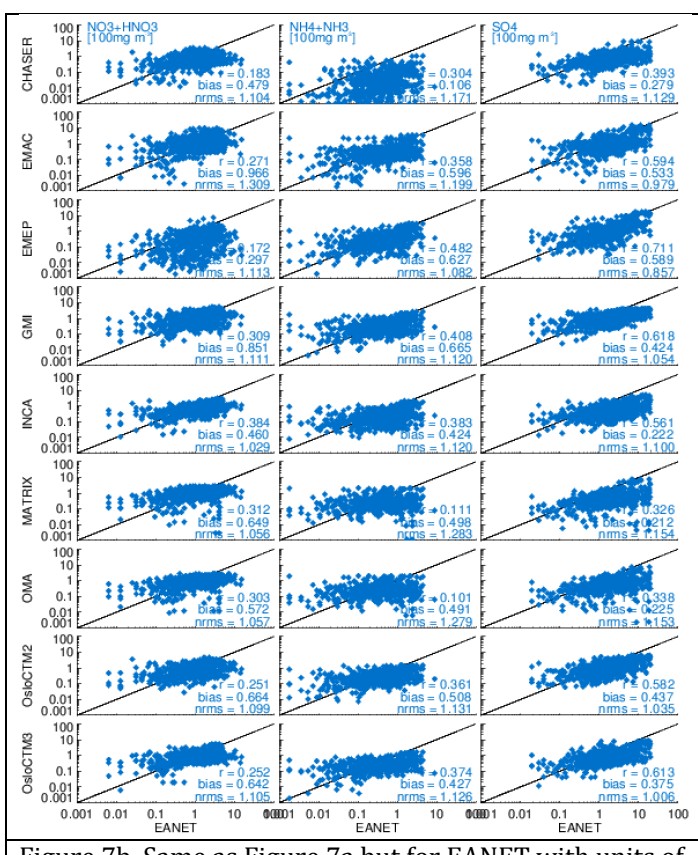

Figure 7b. Same as Figure 7a but for EANET with units of
100mg m$^{-3}$.



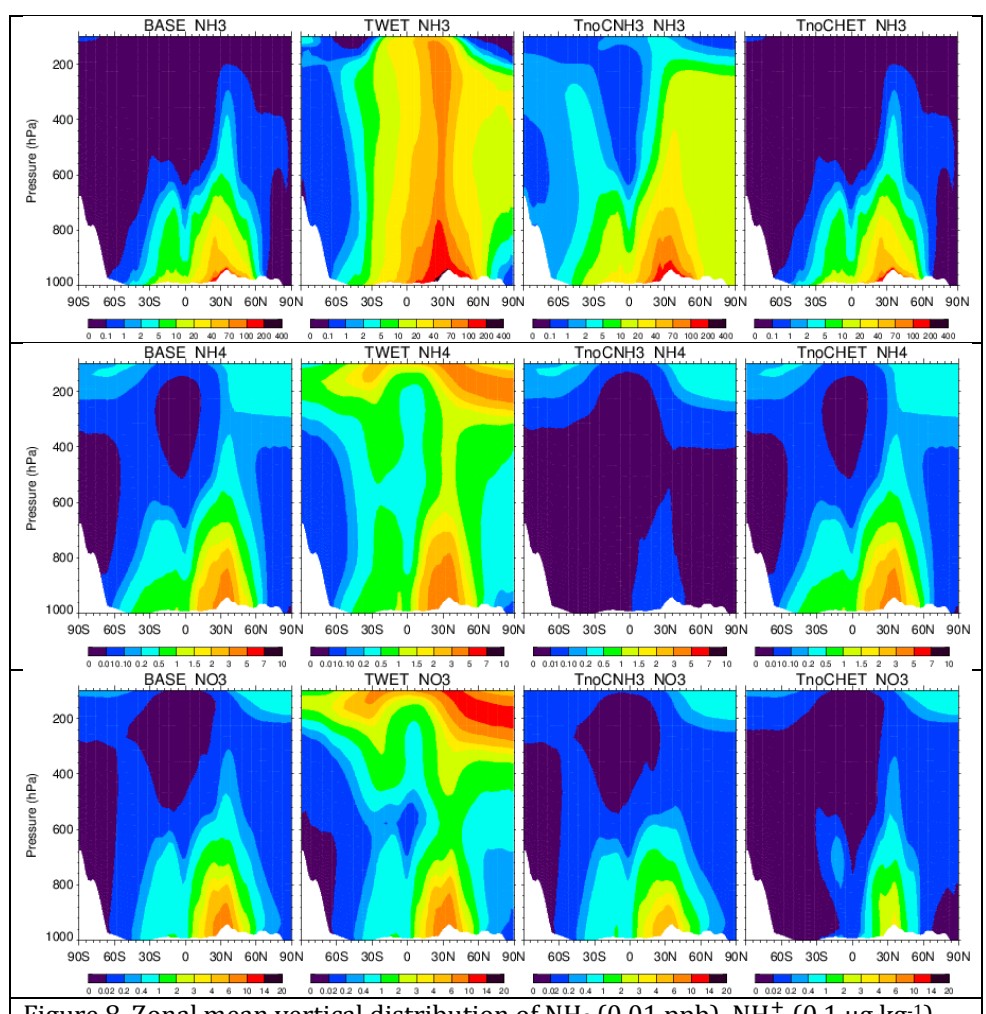

Figure 8. Zonal mean vertical distribution of $NH_3$ (0.01 ppb), $NH_4^+$ (0.1 µg kg$^{-1}$) and $NO_3^-$ (0.05µg kg$^{-1}$) from base and three sensitivity experiments explained in Table 6.





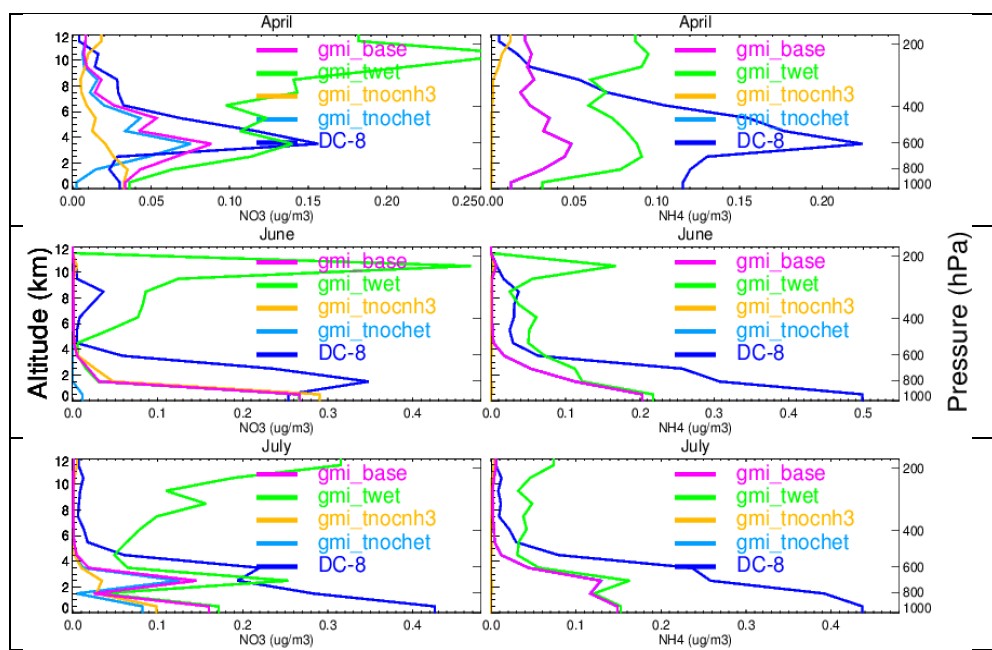

Figure 9. Comparison between GMI simulations and ARCTAS measurements of $NH_4^+$ and $NO_3^-$ from base and three sensitivity experiments explained in Table 6. Note the light blue line for $[NH_4^+]$ is frequently underneath the peak line.

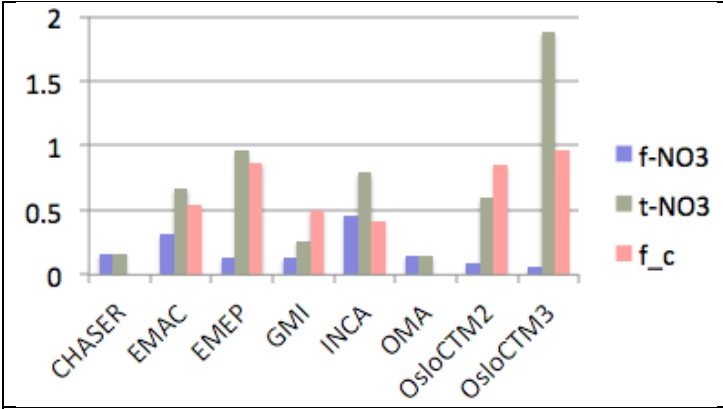

Figure 10. $NO_3^-$ fine mode burden (f-NO3, Tg), total burden (t-NO3, Tg), and coarse mode fraction (f_c) for the eight AeroCom models.