# Peer review of "Investigation of global nitrate from the AeroCom Phase III experiment"

_Atmospheric Chemistry and Physics, 2017_

## Referee Comment (RC1) · Anonymous Referee #1 · 14 Jun 2017

Bian et al. compare global nitrate and ammonium budgets for 9 global chemical models in order to assess differences between the models and attribute these differences to specific processes. This is part of the AeroCom Phase III study. They find that burdens of HNO3 and NO3- differ by factors of 9 and 13, respectively, between the different models. The modeled differences in the NH3/NH4+ burdens were unclear and should be explicitly stated. Modeled chemical production of NH4+ and lifetime differed by factors of 2 and 5, respectively. They attribute these model differences to differences in 1) pH-dependent wet deposition of NH4+, 2) nitrate formation on the surface of sea salt and dust aerosol, and 3) the nitrate coarse mode fraction. They find that nitrate production on sea salt and dust is important to include in models as it tends to dominate nitrate production and controls its partitioning between the fine and coarse mode. In

that sense it seems to me that 2 and 3 above are referring to the same process. They also compare the model results to observations of nitrate and ammonium surface observations of concentrations and deposition, as well as observed vertical profiles from several aircraft campaigns.

Overall this is a well written paper and will be useful for assessing reactive nitrogen budgets in models. One thing I found confusing was the use of the phrase "heterogeneous chemistry" and the use of the term "nitrate". For me, when I hear heterogeneous chemical production of nitrate I think of N2O5 hydrolysis, which this paper did not examine at all. I wonder how nitrate production from N2O5 hydrolysis differs in the models and if this can account for some of the inter-model variability. There was no mention at all of model differences in nitrate production (NO2+OH, BrONO2 hydrolysis, etc) and how this might account for model differences. Perhaps this will be the subject of another paper, and if so it would be nice to mention that here. What the authors are referring to by the use of "heterogeneous chemistry" is what I would call thermodynamic partitioning between the gas and aerosol phase. Perhaps the authors should reconsider their choice of words here so that it is not confusing. Also, when I read "nitrate" I think of HNO3(g) + NO3-, i.e., the sum of gas and particulate nitrate. In this paper, "nitrate" is specifically referring to the particulate phase. Perhaps use the term "particulate nitrate" or "NO3-" instead so it is more clear. That might also partially help with the issue above regarding the term heterogeneous chemistry.

More minor issues:

Lines 363 and 364 need subscripts.

Line 460: replace "decease" with "decrease"

Line 539: What does "the correction of pH in cloud water" mean? It sounds like the models are somehow correcting for a cloud pH calculation. If I understand correctly, it is not the pH calculation that is being corrected, but whether or not pH is being considered in the Henry's law constant calculation for NH3.

[Figure]

Line 569: Check the grammar

Paragraphs beginning on lines 743 and 761 should be combined for clarity.

Line 785: "model" should read "mode"

---

## Referee Comment (RC2) · Anonymous Referee #2 · 4 Jul 2017

This paper presents results from 9 global models with a focus on nitrate aerosol. Since nitrate aerosol formation in linked to ammonia, ammonium, sulfate, and nitric acid, additional species and their deposition is also evaluated. The authors provide insight into the model differences by noting which models include heterogeneous chemistry and pH depending $NH_3$ solubility (Henry's Law). I have one major comment and other minor comments.

Major comment:

At the end of the paper, it is not clear what processes or species future model development should target to improve nitrate aerosol formation. Some insight may be gained by more carefully considering how errors in sulfate (and ammonium) may propagate to errors in aerosol nitrate. In particular, the correlation between model predictions and

observations for NH4 and SO4 is quite poor for some models (Figure 4). Consider Weber et al. (2016) and how decreases in sulfate do not necessarily lead to decreases in aerosol H+ (in contrast to page 2, lines 78-80). As nitrate partitioning is sensitive to pH, nitrate aerosol formation could be limited due to aerosol pH. Weber et al. (2016) and Silvern et al. (2017) have indicated pH may decrease (aerosols become more acidic) in the future. Can the limiting factor (NH3, nitrate, or pH) for nitrate formation be better identified?

Minor comments:

1. The authors should carefully check for awkward wording

2. Line 154: reword to "emission inventories used"

3. Line 186: Was the several months of spinup for meteorology and chemistry or just meterology? Is several months sufficient for chemistry of the upper troposphere?

4. Line 204: Can the differences in organic nitrate treatments be briefly discussed? It would be useful to have production rates of nitric acid from each model.

5. Line 225: Are solid precipitates allowed in any of the metastable configurations?

6. Line 256: typo ISORROPIA-I

7. Line 528-520: sentence is unclear

8. Line 619: Is the goal to compromise accuracy and efficiency?

9. Line 731: Can you clarify what fraction actually used the HTAP2 emission inventory vs something else?

10. Line 753: Do you mean ammonium measured on filters?

11. Line 782: Is it thus possible to recommend that all models use the pH dependent Henry's law coefficient for NH3? Can other recommendations for models be succinctly stated in the conclusions?

12. Table 1: Define CHEMDUSS (not defined until later table)

13. Figure 5: Why are the daily and monthly output results (Figure 5) so different? For the daily output, is the aircraft data matched on a daily basis?

14. Make sure abbreviations are defined in the tables (for example CheAP in 4c and ChemGP in 4d)

References:

Silvern, R. F., Jacob, D. J., Kim, P. S., Marais, E. A., Turner, J. R., Campuzano-Jost, P., and Jimenez, J. L.: Inconsistency of ammonium–sulfate aerosol ratios with thermodynamic models in the eastern US: a possible role of organic aerosol, Atmos. Chem. Phys., 17, 5107-5118, 10.5194/acp-17-5107-2017, 2017.

Weber, R. J., Guo, H., Russell, A. G., and Nenes, A.: High aerosol acidity despite declining atmospheric sulfate concentrations over the past 15 years, Nature Geosci., 9, 282-285, 10.1038/ngeo2665, 2016.

---

## Author Comment (AC1) · 19 Jul 2017

We thank the instructive and detail points given by the two reviewers. We have carefully revised our manuscript accordingly. Below is our point-to-point response.

Reviewer1:

Reviewer: Bian et al. compare global nitrate and ammonium budgets for 9 global chemical models in order to assess differences between the models and attribute these differences to specific processes. This is part of the AeroCom Phase III study. They find that burdens of HNO3 and NO3- differ by factors of 9 and 13, respectively, between the different models. The modeled differences in the NH3/NH4+ burdens were unclear and should be explicitly stated. Modeled chemical production of NH4+ and lifetime differed by factors of 2 and 5, respectively. They attribute these model differences to differences in 1) pH-dependent wet deposition of NH4+, 2) nitrate formation on the surface of sea salt and dust aerosol, and 3) the nitrate coarse mode fraction. They find that nitrate production on sea salt and dust is important to include in models as it tends to dominate nitrate production and controls its partitioning between the fine and coarse mode. In that sense it seems to me that 2 and 3 above are referring to the same process. They also compare the model results to observations of nitrate and ammonium surface observations of concentrations and deposition, as well as observed vertical profiles from several aircraft campaigns.

Authors: A sentence that describes the differences in the NH3/NH4+ burdens is added right after the description for HNO3 and NO3 in the abstract (lines 38-39). We intend to separate discussion of section 5.2 and 5.3 because the nitrate formation on the surface of sea salt and dust aerosol (section 5.3) is important, but not the only factor, to determining nitrate size distribution (section 5.2). Also, the former focuses more on chemical process and the later on physical process and climate implication.

Reviewer: Overall this is a well written paper and will be useful for assessing reactive nitrogen budgets in models. One thing I found confusing was the use of the phrase "heterogeneous chemistry" and the use of the term "nitrate". For me, when I hear heterogeneous chemical production of nitrate I think of N2O5 hydrolysis, which this paper did not examine at all. I wonder how nitrate production from N2O5 hydrolysis differs in the models and if this can account for some of the inter-model variability. There was no mention at all of model differences in nitrate production (NO2+OH, BrONO2 hydrolysis, etc) and how this might account for model differences. Perhaps this will be the subject of another paper, and if so it would be nice to mention that here. What the authors are referring to by the use of "heterogeneous chemistry" is what I would call thermodynamic partitioning between the gas and aerosol phase. Perhaps the authors should reconsider their choice of words here so that it is not confusing. Also, when I read "nitrate" I think of HNO3(g) + NO3-, i.e., the sum of gas and particulate nitrate. In

this paper, "nitrate" is specifically referring to the particulate phase. Perhaps use the term "particulate nitrate" or "NO3-" instead so it is more clear. That might also partially help with the issue above regarding the term heterogeneous chemistry.

Authors: We add a sentence "hereafter nitrate referring to particulate nitrate unless otherwise specified" in the introduction (line 56). We also add the following clarification in section 2.2 (lines 216-220). "Please also note that the heterogeneous chemical production of nitrate mentioned in this paper refers only the reaction of HNO3 on dust and sea salt particles. A series of reactions, such as N2O5 hydrolysis and BrONO2 hydrolysis, affect HNO3 simulation. These reactions are typically considered in O3-NOx-HOx chemistry and their discussion is beyond the scope of this paper."

More minor issues: Lines 363 and 364 need subscripts. Authors: Done.

Line 460: replace "decease" with "decrease" Authors: Done.

Line 539: What does "the correction of pH in cloud water" mean? It sounds like the models are somehow correcting for a cloud pH calculation. If I understand correctly, it is not the pH calculation that is being corrected, but whether or not pH is being considered in the Henry's law constant calculation for NH3. Authors: Delete "correction of" before pH.

Line 569: Check the grammar Authors: The sentence has been revised to be "The latter corresponds to a range of pH from 4.5 (Oslo-CTM2) to 5.5 (CHASER)."

Paragraphs beginning on lines 743 and 761 should be combined for clarity. Authors: Yes, combined now.

Line 785: "model" should read "mode" Authors: Done.

Reviewer2:

Reviewer: This paper presents results from 9 global models with a focus on nitrate aerosol. Since nitrate aerosol formation in linked to ammonia, ammonium, sulfate, and nitric acid, additional species and their deposition is also evaluated. The authors provide insight into the model differences by noting which models include heterogeneous chemistry and pH depending NH3 solubility (Henry's Law). I have one major comment and other minor comments.

Major comment: At the end of the paper, it is not clear what processes or species future model development should target to improve nitrate aerosol formation. Some insight may be gained by more carefully considering how errors in sulfate (and ammonium) may propagate to errors in aerosol nitrate. In particular, the correlation between model predictions and observations for NH4 and SO4 is quite poor for some models (Figure 4). Consider Weber et al. (2016) and how decreases in sulfate do not necessarily lead to decreases in aerosol H+ (in contrast to page 2, lines 78-80). As nitrate partitioning is sensitive to pH, nitrate aerosol formation could be limited due to aerosol pH. Weber et al. (2016) and Silvern et al. (2017) have indicated pH may decrease (aerosols become more acidic) in the future. Can the limiting factor (NH3, nitrate, or pH) for nitrate formation be better identified? Authors: Thanks to the reviewer for bringing this insightful point to the discussion of potential future study. We have expanded the discussion in the conclusion (lines 814-839). "Our work presents a first effort to assess nitrate simulation from chemical and physical processes. A companion study is proposed by AeroCom III nitrate activity to investigate how sensitive is nitrate formation in response to the possible future changes of emission and meteorological fields. These perturbation fields include increasing NH3 emission, decreasing NOx, SOx and dust emissions, and increasing atmospheric temperature and relative humidity. It would be particularly interesting to examine how aerosol pH changes and its influence on atmospheric acid/base gas-particle system during the experiment. Future aerosol pH does not necessarily increase with SO2 emission reduction. Indeed, studies over US southeast indicated that its aerosol has been getting more acidic over the past decade although SO2 emission decreased and NH3 emission stayed constant [Silvern et al., 2017; Weber et al., 2016]. This environment of high aerosol acidity hinders the formation of nitrate aerosol, which only occurs when pH is over ∼2 to 3 [Weber et al.,

2016]. In addition, understanding why and how the system is insensitive to changing SO2 level due to buffering of the partitioning of semivolatile NH3 over regions such as US southeast helps us to gain some insight into how errors in sulfate (and ammonium) may propagate to errors in aerosol nitrate. In particular, the correlation between model predictions and observations for SO_4ˆ(2-)and ãĂŰNHãĂŮ_4ˆ+ is quite poor for some models (Figure 4). It would be also interesting to include organic gas/aerosol into the system since they are not only important atmospheric components, but also reduce the uptake of NH3. Competition for uptake between NH3 and organic gases considerably slows down the approach to thermodynamic equilibrium [Silvern et al., 2017]. Based on the findings of this work, modelers should pay particular attention to incorporating dust and sea salt and treating NH3 wet deposition to improve nitrate simulation. Further evaluation using satellite measurements, such as NH3 products from IASI and TES, is desired and will be conducted. Such evaluation requires global 3-dimensional high frequency model data. Potential future study also includes estimation of nitrate forcing for climate change."

By the way, the sentence in original submission page 2 lines 78-80 just states the facts of abundant NO3 and SO4 observed in atmosphere.

Minor comments: 1. The authors should carefully check for awkward wording Authors: Yes. We have also revised the manuscript based at the reviewers' suggestion.

2. Line 154: reword to "emission inventories used" Authors: Done.

3. Line 186: Was the several months of spinup for meteorology and chemistry or just meterology? Is several months sufficient for chemistry of the upper troposphere? Authors: The spinup period is for chemistry simulation. We have changed "several months" to "one-year" as specified in the protocol of AeroCom III nitrate experiment. One-year of spinup should be fine for the chemical species discussed in this study in the upper troposphere.

4. Line 204: Can the differences in organic nitrate treatments be briefly discussed? It would be useful to have production rates of nitric acid from each model. Authors: Although gas- and aerosol-phase organic nitrates are important, the chemistry formation and degradation remains uncertain [Fisher et al., 2016]. To my knowledge, the models involved in this experiment do not have aerosol-phase organic nitrate. We agree with the reviewer on the usefulness of having production rates of nitric acid from each model. We specified this requirement in the experiment protocol. Unfortunately only two models submitted this kind of data, which were presented in our table 4c.

Fisher, J. A., Jacob, D. J., Travis, K. R., Kim, P. S., Marais, E. A., Chan Miller, C., Yu, K., Zhu, L., Yantosca, R. M., Sulprizio, M. P., Mao, J., Wennberg, P. O., Crounse, J. D., Teng, A. P., Nguyen, T. B., St. Clair, J. M., Cohen, R. C., Romer, P., Nault, B. A., Wooldridge, P. J., Jimenez, J. L., Campuzano-Jost, P., Day, D. A., Hu, W., Shepson, P. B., Xiong, F., Blake, D. R., Goldstein, A. H., Misztal, P. K., Hanisco, T. F., Wolfe, G. M., Ryerson, T. B., Wisthaler, A., and Mikoviny, T.: Organic nitrate chemistry and its implications for nitrogen budgets in an isoprene- and monoterpene-rich atmosphere: constraints from aircraft (SEAC4RS) and ground-based (SOAS) observations in the Southeast US, Atmos. Chem. Phys., 16, 5969-5991, https://doi.org/10.5194/acp-16-5969-2016, 2016.

5. Line 225: Are solid precipitates allowed in any of the metastable configurations? Authors: No. For a metastable configuration, aerosol is composed only of an aqueous phase that can be supersaturated with respect to dissolved salts.

6. Line 256: typo ISORROPIA-I Authors: Done.

7. Line 528-520: sentence is unclear Authors: Change the sentence to "Consequently, the slopes of the fitting lines are generally less than 1 on the scattering plots with model as y-axis and observation as x-axis (e.g. Figures 4a-d, 6, 7a-b)."

8. Line 619: Is the goal to compromise accuracy and efficiency? Authors: Yes.

9. Line 731: Can you clarify what fraction actually used the HTAP2 emission inventory vs something else? Authors: The detailed discussion for the fraction actually used the HTAP2 emission inventory is presented in section 2.1. We have added "for aerosol and ozone simulations" after "Emissions from anthropogenic, aircraft, and ship" on line 171. We have also added "for the aerosol and gas emissions from anthropogenic, aircraft, and ship sources" after "use HTAP2 emission inventory" on lines 738-739.

10. Line 753: Do you mean ammonium measured on filters? Authors: Yes.

11. Line 782: Is it thus possible to recommend that all models use the pH dependent Henry's law coefficient for NH3? Can other recommendations for models be succinctly stated in the conclusions? Authors: Since liquid-phase reaction 2 in Appendix can reach equilibrium quickly within a chemical time step, we recommend including it in accounting for NH3 solution. Theoretically, a more accurate approach is to combine wet removal with liquid-phase chemistry calculation. In other words, instead of using an implicit calculation of the effective Henry's law constant, the gas-liquid phase equilibrium is explicitly calculated based on the chemical mechanism used in the liquid phase. The solution of NH3 is calculated by solving a set of partial differential equations, which includes not only the gas-liquid phase equilibrium, but also all the important reactions in the liquid phase, as adopted in EMAC model. We have added this discussion in the conclusion (lines 788-796).

12. Table 1: Define CHEMDUSS (not defined until later table) Authors: Done

13. Figure 5: Why are the daily and monthly output results (Figure 5) so different? For the daily output, is the aircraft data matched on a daily basis? Authors: The big difference between the daily and monthly output is mainly shown by the ATCTAS April campaign. The April experiment was conducted over Alaska for long-range transport of Asia pollution so that the day-to-day atmospheric dynamic variation could play more on the pollution over Alaska. For the daily output, the model and aircraft data match on a daily basis.

14. Make sure abbreviations are defined in the tables (for example CheAP in 4c and

ChemGP in 4d) Authors: Done.

We thank the instructive and detail points given by the two reviewers. We have carefully revised our manuscript accordingly. Below is our point-to-point response.

Reviewer1:

Reviewer: Bian et al. compare global nitrate and ammonium budgets for 9 global chemical models in order to assess differences between the models and attribute these differences to specific processes. This is part of the AeroCom Phase III study. They find that burdens of HNO3 and NO3- differ by factors of 9 and 13, respectively, between the different models. The modeled differences in the NH3/NH4+ burdens were unclear and should be explicitly stated. Modeled chemical production of NH4+ and lifetime differed by factors of 2 and 5, respectively. They attribute these model differences to differences in 1) pH-dependent wet deposition of NH4+, 2) nitrate formation on the surface of sea salt and dust aerosol, and 3) the nitrate coarse mode fraction. They find that nitrate production on sea salt and dust is important to include in models as it tends to dominate nitrate production and controls its partitioning between the fine and coarse mode. In that sense it seems to me that 2 and 3 above are referring to the same process. They also compare the model results to observations of nitrate and ammonium surface observations of concentrations and deposition, as well as observed vertical profiles from several aircraft campaigns.

**Authors: A sentence that describes the differences in the NH3/NH4+ burdens is added right after the description for HNO3 and NO3 in the abstract (lines 38-39). We intend to separate discussion of section 5.2 and 5.3 because the nitrate formation on the surface of sea salt and dust aerosol (section 5.3) is important, but not the only factor, to determining nitrate size distribution (section 5.2). Also, the former focuses more on chemical process and the later on physical process and climate implication.**

Reviewer: Overall this is a well written paper and will be useful for assessing reactive nitrogen budgets in models. One thing I found confusing was the use of the phrase "heterogeneous chemistry" and the use of the term "nitrate". For me, when I hear heterogeneous chemical production of nitrate I think of N2O5 hydrolysis, which this paper did not examine at all. I wonder how nitrate production from N2O5 hydrolysis differs in the models and if this can account for some of the inter-model variability. There was no mention at all of model differences in nitrate production (NO2+OH, BrONO2 hydrolysis, etc) and how this might account for model differences. Perhaps this will be the subject of another paper, and if so it would be nice to mention that here. What the authors are referring to by the use of "heterogeneous chemistry" is what I would call thermodynamic partitioning between the gas and aerosol phase. Perhaps the authors should reconsider their choice of words here so that it is not confusing. Also, when I read "nitrate" I think of HNO3(g) + NO3-, i.e., the sum of gas and particulate nitrate. In this paper, "nitrate" is specifically referring to the particulate phase. Perhaps use the term "particulate nitrate" or "NO3-" instead so it is more clear. That might also partially help with the issue above regarding the term heterogeneous chemistry.

**Authors: We add a sentence "hereafter nitrate referring to particulate nitrate unless otherwise specified" in the introduction (line 56). We also add the following clarification in section 2.2 (lines 216-220). "Please also note that the heterogeneous chemical production of nitrate mentioned in this paper refers only the reaction of HNO$_3$ on dust and sea salt particles. A series of reactions, such as N$_2$O$_5$ hydrolysis and BrONO$_2$ hydrolysis, affect**

**Fig. 1.**

**Investigation of global nitrate from the AeroCom Phase III experiment**
Huisheng Bian[1,2], Mian Chin[2], Didier A. Hauglustaine[3], Michael Schulz[4], Gunnar Myhre[5,6],
Susanne E. Bauer[7,8], Marianne T. Lund[6], Vlassis A. Karydis[9], Tom L. Kucsera[10], Xiaohua Pan[11],
Andrea Pozzer[9], Ragnhild B. Skeie[6], Stephen D. Steenrod[10], Kengo Sudo[12], Kostas
Tsigaridis[7,8], Alexandra P. Tsimpidi[9], and Svetlana G. Tsyro[4]
[1] Joint Center for Environmental Technology UMBC, Baltimore, MD, USA
[2] Laboratory for Atmospheres, NASA Goddard Space Flight Center, Greenbelt, MD, USA
[3] Laboratoire des Sciences du Climat et de l'Environnement (LSCE), UMR8212, CEA-CNRS-UVSQ, Gif-
sur-Yvette, France
[4] Norwegian Meteorological Institute, Blindern, Norway
[5] Department of Geosciences, University of Oslo, Oslo, Norway
[6] Center for International Climate and Environmental Research-Oslo, Oslo, Norway
[7] The Earth Institute, Center for Climate Systems Research, Columbia University, New York, USA
[8] NASA Goddard Institute for Space Studies, New York, USA
[9] Max Planck Institute for Chemistry, 55128 Mainz, Germany
[10] Universities Space Research Association, GESTAR, Columbia, MD, USA
[11] School of Computer, Mathematical and Natural Sciences, Morgan State University, Baltimore, MD, USA
[12] Center for Climate System Research, University of Tokyo, Tokyo, Japan.
**Abstract**
An assessment of global particulate nitrate and ammonium aerosol based on simulations
from nine models participating in the AeroCom Phase III study is presented. A budget
analyses was conducted to understand the typical magnitude, distribution, and diversity
of the aerosols and their precursors among the models. To gain confidence on model
performance, the model results were evaluated with various observations globally,
including ground station measurements over North America, Europe, and East Asia for
tracer concentrations and dry and wet depositions, as well as with aircraft measurements
in the Northern Hemisphere mid-high latitudes for tracer vertical distributions. Given the
unique chemical and physical features of the nitrate occurrence, we further investigated
the similarity and differentiation among the models by examining: 1) the pH-dependent
$NH_3$ wet deposition; 2) the nitrate formation via heterogeneous chemistry on the surface
of dust and sea-salt particles; and 3) the nitrate coarse mode fraction (i.e., coarse/total). It
is found that $HNO_3$, which is simulated explicitly based on full $O_3$-$HO_x$-$NO_x$-aerosol
chemistry by all models, differs by up to a factor of 9 among the models in its global
tropospheric burden. This partially contributes to a large difference in $NO_3^-$, whose
atmospheric burden differs by up to a factor of 13. The atmospheric burdens of $NH_3$ and
$NH_4^+$ differ by 17 and 4, respectively. Analyses at the process level show that the large
diversity in atmospheric burdens of $NO_3^-$, $NH_3$, and $NH_4^+$ is also related to deposition
processes. Wet deposition seems to be the dominant process in determining the diversity
in $NH_3$ and $NH_4^+$ lifetimes. It is critical to correctly account for contributions of
heterogeneous chemical production of nitrate on dust and sea-salt, because this process
overwhelmingly controls atmospheric nitrate production (typically >80%) and determines
the coarse and fine mode distribution of nitrate aerosol.
**1. Introduction**
Atmospheric aerosols adversely affect human health and play an important role in
changing the Earth's climate. A series of multimodel studies have been coordinated by

[Figure]

**Fig. 2.**

---

## Author Comment (AC2) · 22 Aug 2017

[revised manuscript text omitted]

concentrations due to the gas/particle equilibrium). Meanwhile, all models consider $N_2O_5$
hydrolysis, the conversion of $N_2O_5$ to $HNO_3$. The first order loss reaction occurs on the
surface of tropospheric aerosols and assumes irreversible instant reaction.  However, the
models differ in $N_2O_5$ hydrolysis by considering the reaction on the surface of different
aerosol types. Uptake coefficients (aka gamma factors) also differ in their relationship to
temperature and RH. CHASER model is special as it allows $N_2O_5$ conversion to $HNO_3$
on liquid cloud particles. Please refer to Table 1 and the listed references for details. 
[revised manuscript text omitted]
 coarse mode. Coarse mode aerosol nitrate is formed due to presence of dust and/or sea salt. In other words, the
formation of nitrate on coarse mode dust and sea salt particles is the major factor
controlling size distribution. Other factors, such as $NH_3/NH_4^+/NO_3^-$ chemistry and
atmospheric transport and removal processes, also affect nitrate size distribution. Having
an accurate aerosol size distribution is critical in climate forcing estimations, since large
size particles have a relatively small optical cross section at a given aerosol mass loading
and the nitrate material coating on dust particles has almost no direct impact on the dust
optics, although the greatly impact dust lifetime (Bauer et al., 2007). Given that the
deposition velocity of a coarse particle is greater than that of a fine particle, an accurate
size distribution is also necessary to estimate deposition of particulate nitrates (Yeatman
et al., 2001; Sadanaga et al., 2008). This estimation is particularly important over oceans
where coarse mode nitrate dominates (Itahashi et al., 2016) and nitrogen supply is often
in deficit (Hansell and Follows, 2008).
As we have discussed in section 5.2, nitrate size distribution varies with the approaches
adopted for nitrate formation on coarse mode aerosols (i.e. dust and sea salt). Figure 10
gives the burdens of nitrate in fine mode and coarse mode portions and the ratio between
coarse mode and total (f_c) for the eight discussed models. The ratio is ranging from 0
(CHASER and GISS-OMA), ~50% (EMAC, GMI and INCA), ~80% (EMEP and
OsloCTM2), and 97% (OsloCTM3). The two OsloCTMs give the highest f_c partially
because they run TEQM model for coarse model particles.
A wide range of f_c, from 0 to > 90%, has been reported previously by model
simulations (Adams et al., 2001; Bauer et al., 2007; Jacobson 2001), while the range is
narrowed down to 40-60% for the model studies using the approach that solves dynamic
mass transfer equation for coarse mode particles (Feng and Penner, 2007; Xu and Penner,
2012).
It is worth pointing out that aerosol microphysics modify aerosol size as well. For
example, a process like coagulation would also allow $NO_3^-$ to mix with other particles and
enter coarse mode aerosol. New particle formation/nucleation would add $NH_3/NH_4^+/NO_3^-$
into the ultra fine mode. Except EMAC and GISS-MATRIX, majority models involved in
this study are bulk aerosol models that do not account for aerosol microphysics.
It is challenging to verify the nitrate size distribution globally due to the limited
measurements on time and space. Measurements over regional and station sites indicated
that the ratio of f_c could be very high and vary seasonally over oceanic sites. For
example, annual mean f_c during 2002-2004 from the Fukue supersite observatory is
about 72% with a seasonal variation of 60–80% in winter and of around 80% in summer
(Itahashi et al., 2016).
However, the ratio could be varied dramatically over land or the areas affected by land
pollution. For example, observations of fine and coarse particulate nitrate at several rural
locations in the United States indicated that nitrate was predominantly in submicron
ammonium nitrate particles during the Bondville and San Gorgonio (April) campaigns, in
coarse mode nitrate particles at Grand Canyon (May) and Great Smoky Mountains (July/August), and both fine and coarse mode nitrate during the studies at Brigantine and
San Gorgonio (July) (Lee et al., 2008). Allen et al. (2015) examined aerosol composition
data collected during the summer 2013 SOAS and concluded that inorganic nitrate in the
southeastern United States likely exists in the form of supermicron $NO_3^-$, balanced by the
presence of mineral cations arising from the transport of crustal dust and sea spray
aerosol. The measurements over Harvard Forest, a rural site in central Massachusetts,
supported that the majority of nitrate mass was associated with water-soluble
supermicron soil-derived $Ca^{2+}$ in an acidic environment (Lefer and Talbot, 2001).
Measurements taken in Paris during the ESQUIF campaign found that the coarse
nitrate fraction represents up to 60% of total particulate nitrate mass at night and 80% at

[revised manuscript text omitted]

Søvde, O. A., Prather, M. J., Isaksen, I. S. A., Berntsen, T. K., Stordal, F., Zhu, X., Holmes, C. D., and Hsu,
J.: The chemical transport model Oslo CTM3, Geosci. Model Dev., 5, 1441-1469,
https://doi.org/10.5194/gmd-5-1441-2012, 2012.
Strahan, S. E., Duncan, B. N., and Hoor, P.: Observationally derived transport diagnostics for the
lowermost stratosphere and their application to the GMI chemistry and transport model, Atmos. Chem.
Phys., 7, 2435-2445, doi:10.5194/acp-7-2435-2007, 2007.
Sudo, K., M. Takahashi, J. Kurokawa, and H. Akimoto, CHASER: A global chemical model of the
troposphere, 1. Model description, J. Geophys. Res., 107(D17), 4339, doi:10.1029/2001JD001113,
2002.
Takiguchi, Y., A. Takami, Y. Sadanaga, X. Lun, A. Shimizu, I. Matsui, N. Sugimoto, W. Wang, H.
Bandow, and S. Hatakeyama (2008), Transport and transformation of total reactive nitrogen over the
East China Sea, J. Geophys. Res., 113, D10306, doi:10.1029/2007JD009462.
Textor, C., Schulz, M., Guibert, S., Kinne, S., Balkanski, Y., Bauer, S., Berntsen, T., Berglen, T., Boucher,
O., Chin, M., Dentener, F., Diehl, T., Easter, R., Feichter, H., Fillmore, D., Ghan, S., Ginoux, P.,
Gong, S., Grini, A., Hendricks, J., Horowitz, L., Huang, P., Isaksen, I., Iversen, I., Kloster, S., Koch,
D., Kirkevåg, A., Kristjansson, J. E., Krol, M., Lauer, A., Lamarque, J. F., Liu, X., Montanaro, V.,
Myhre, G., Penner, J., Pitari, G., Reddy, S., Seland, Ø., Stier, P., Takemura, T., and Tie, X.: Analysis and quantification of the diversities of aerosol life cycles within AeroCom, Atmos. Chem. Phys., 6,
1777–1813, doi:10.5194/acp-6-1777-2006, 2006.
Tost, H., Jöckel, P., Kerkweg, A., Sander, R., and Lelieveld, J.: Technical note: A new comprehensive
SCAVenging submodel for global atmospheric chemistry modelling, Atmos. Chem. Phys., 6, 565-574,
doi:10.5194/acp-6-565-2006, 2006.
Trail, M., Tsimpidi, A. P., Liu, P., Tsigaridis, K., Rudokas, J., Miller, P., Nenes, A., Hu, Y., and Russell, A.
G.: Sensitivity of air quality to potential future climate change and emissions in the United States and
major cities, Atmospheric Environment, 94, 552-563, 2014.
Trump, E. R., Fountoukis, C., Donahue, N. M., and Pandis, S. N.: Improvement of simulation of fine
inorganic PM levels through better descriptions of coarse particle chemistry, Atmospheric
Environment, 102, 274-281, 2015.
Tsigaridis, K., Daskalakis, N., Kanakidou, M., Adams, P. J., Artaxo, P., Bahadur, R., Balkanski, Y.,
Bauer, S. E., Bellouin, N., Benedetti, A., Bergman, T., Berntsen, T. K., Beukes, J. P., Bian, H.,
Carslaw, K. S., Chin, M., Curci, G., Diehl, T., Easter, R. C., Ghan, S. J., Gong, S. L., Hodzic, A.,
Hoyle, C. R., Iversen, T., Jathar, S., Jimenez, J. L., Kaiser, J. W., Kirkevåg, A., Koch, D., Kokkola, H.,
Lee, Y. H, Lin, G., Liu, X., Luo, G., Ma, X., Mann, G. W., Mihalopoulos, N., Morcrette, J.-J.,
Müller, J.-F., Myhre, G., Myriokefalitakis, S., Ng, N. L., O'Donnell, D., Penner, J. E., Pozzoli, L.,
Pringle, K. J., Russell, L. M., Schulz, M., Sciare, J., Seland, Ø., Shindell, D. T., Sillman, S.,
Skeie, R. B., Spracklen, D., Stavrakou, T., Steenrod, S. D., Takemura, T., Tiitta, P., Tilmes, S.,
Tost, H., van Noije, T., van Zyl, P. G., von Salzen, K., Yu, F., Wang, Z., Wang, Z., Zaveri, R. A.,
Zhang, H., Zhang, K., Zhang, Q., and Zhang, X.: The AeroCom evaluation and intercomparison of
organic aerosol in global models, Atmos. Chem. Phys., 14, 10845-10895, doi:10.5194/acp-14-10845-
2014, 2014.
Tsimpidi, A. P., Karydis, V. A., and Pandis, S. N.: Response of inorganic fine particulate matter to
emission changes of sulfur dioxide and ammonia: The eastern United States as a case study, Journal of
the Air & Waste Management Association, 57, 1489-1498, 2007.
Tsimpidi, A. P., Karydis, V. A., and Pandis, S. N.: Response of Fine Particulate Matter to Emission
Changes of Oxides of Nitrogen and-Anthropogenic Volatile Organic Compounds in the Eastern United
States, Journal of the Air & Waste Management Association, 58, 1463-1473, 2008.
Vieno, M., Heal, M. R., Twigg, M. M., MacKenzie, I. A., Braban, C. F., Lingard, J. J. N. Ritchie, S., Beck,
R. C., A., M., Ots, R., DiMarco, C. F., Nemitz, E., Sutton, M. A., and Reis, S.: The UK particulate
matter air pollution episode of March-April 2014: more than Saharan dust., Environ. Res. Lett.,
doi:10.1088/1748-9326/11/4/044004, 2016.
Walker, J. M., Philip, S., Martin, R. V., and Seinfeld, J. H.: Simulation of nitrate, sulfate, and ammonium
aerosols over the United States, Atmos. Chem. Phys., 12, 11213–11227, doi:10.5194/acp-12-11213-
2012, 2012.
Watanabe, S., T. Hajima, K. Sudo, T. Nagashima, T. Takemura, H. Okajima, *et al*. MIROC-ESM 2010:
model description and basic results of CMIP5-20c3m experiments, Geosci. Model Dev., 4 (2011), pp.
845–872.
Weber, R. J., Guo, H., Russell, A. G., and Nenes, A.: High aerosol acidity despite declining
atmospheric sulfate concentrations over the past 15 years, Nature Geosci., 9, 282-285,
10.1038/ngeo2665, 2016.
Williams, E. J., S. T. Sandholm, J. D. Bradshaw, J. S. Schendel, A. O. Langford, P. K. Quinn, P. J. LeBel,
S. A. Vay, P. D. Roberts, R. B. Norton, B. A. Watkins, M. P. Buhr, D. D. Parrish, J. G. Calvert, and F.
C. Fehsenfeld, An intercomparison of five ammonia measurement techniques, J. Geophys. Res., Vol.,
97, No. D11, Pages 11591-11611, July 20, 1992.
van der Werf, G. R., Randerson, J. T., Giglio, L., Collatz, G. J., Mu, M., Kasibhatla, P. S., Morton, D. C.,
DeFries, R. S., Jin, Y., and van Leeuwen, T. T.: Global fire emissions and the contribution of
deforestation, savanna, forest, agricultural, and peat fires (1997–2009), Atmos. Chem. Phys., 10,
11707-11735, doi:10.5194/acp-10-11707-2010, 2010.
Zender, C. S., Bian, H. S., and Newman, D.: Mineral Dust Entrainment and Deposition (DEAD) model:
Description and 1990s dust climatology, J. Geophys. Res.-Atmos., 108, 4416,
doi:10.1029/2002jd002775, 2003.
Zhou, J., B. Gu, W. H. Schlesinger, X. Ju, Significant accumulation of nitrate in Chinese semi-humid
croplands, Scientific Reports 6, Article number: 25088, doi:10.1038/srep25088, 2016.

**Table 1. Nitrate chemical mechanism and physical properties of AeroCom models**

| Model | CHEM-EQM | HNO₃ chem mechanism | N₂O₅ Hydrolysis | CHEM DUST | CHEM SEASALT | How do CHEMDUSS[a] | Bins for nitrate | Model Name & resolution | References |
|---|---|---|---|---|---|---|---|---|---|
| CHASER | ISORROPIA-I | CHASER (Sudo et al., 2002) | $\gamma$[b] (0.1 for $SO_4^{2-}$, $NO_3^-$, OC, DU, and SS, and 0.05 for liquid cloud particles) (Dentener and Crutzen, 1993) | No | No | --- | Fine mode | MIROC, GCM, 2.8°x2.8°x64 | Watanabe et al., 2011 |
| EMAC | ISORROPIA-II (Stable state[c]) | MESSy2 (Jöckel et al., 2010) | $\gamma$ (STA), STA[d]: climatological aerosol in Aitken, accumulation, and coarse soluble modes (Jöckel et al 2010). | Yes | Yes | ISORROPIA-II (TEQM) | 4 bins: Nucleation, Aitken, Accumulation, Coarse | ECHAM5, GCM, 2.8°x2.8°x31 | Karydis et al., 2016 |
| EMEP | MARS | EMEP EmChem09 (Simpson et al., 2012) | $\gamma$ (STA, T, RH), STA: $NH_4^+$, $SO_4^{2-}$, $NO_3^-$ (Evans and Jacob, 2005; Davis et al.,2008) | Yes | Yes | First order loss (HETCHEM) | Fine and coarse | ECMWF-IFS, CTM, 0.5x0.5°x20 | Simpson et al., 2012 |
| GMI | RPMARES (Stable state) | GMI (Straham et al., 2007) | $\gamma$(STA, T, RH), STA: BC, OC, $SO_4^{2-}$, DU, SS (Evans and Jacob, 2005). | Yes | Yes | first order loss (HETCHEM) | 3 bins: (D<0.1, 0.1 – 2.5, > 2.5 um) | MERRA2, CTM, 2.5°x2°x72 | Bian et al., 2009 |
| INCA | INCA (Stable state) | INCA tropospheric chemistry (Hauglustaine et al., 2004) | $\gamma$(STA, T, RH), STA: BC, $SO_4^{2-}$, DU, SS (Evans and Jacob, 2005). | Yes | Yes | first order loss (HETCHEM) | 2 bins : (D< 1µm and 1 - 10µm) | LMD-v4, GCM, 1.9°x3.75°x39 | Hauglustaine et al., 2014 |
| GISS MATRIX | ISORROPIA-II (Stable state) | MATRIX Bauer (2008) and tropospheric chemistry (Shindell et al., 2003) | $\gamma$ (STA), STA: $SO_4^{2-}$ (Dentener and Crutzen, 1993) | No | No | NO | Distributed over all mixing states e.g. size distributions. | NASA GISS-E2, GCM, 2°x2.5°x40 | Schmidt et al 2014 |
| GISS OMA | EQSAM_v03d (Metastable[e]) | OMA (Bauer 2007) and tropospheric chemistry (Shindell et al., 2003) | $\gamma$ (STA), STA: $SO_4^{2-}$ (Dentener and Crutzen, 1993) | Yes | No | Bauer and Koch, 2005 (HETCHEM) | Fine mode | NASA GISS-E2, GCM, 2°x2.5°x40 | Schmidt et al 2014 |
| Oslo CTM2 | EQSAM_v03d (Metastable) | Oslo CTM2 (Berntsen and Isaksen, 1997) | $\gamma$ (STA), STA: climatology aerosol (Dentener and Crutzen, 1993; Søvde et al., 2012). | No | Yes | EQSAM_v03d (TEQM) | 2 bins: fine and coarse mode | ECMWF, CTM, 2.8°x2.8°x60 | Myhre et al., 2006 |
| Oslo CTM3 | EQSAM_v03d (Metastable) | Oslo CTM2 (Berntsen and Isaksen, 1997) | $\gamma$ (STA), STA: climatology aerosol (Dentener and Crutzen, 1993; Søvde et al., 2012). | No | Yes | EQSAM_v03d (TEQM) | 2 
[revised manuscript text omitted]

---

## Editor Decision (ED1)

I thank authors for responding to both reviewers. I have some comments regarding their responses and some additional remarks that need to be considered before publication.

**Concerning responses to R1:**

R1 comment #1: "They attribute these model differences to differences in 1) pH-dependent wet deposition of NH4+, 2) nitrate formation on the surface of sea salt and dust aerosol, and 3) the nitrate coarse mode fraction. They find that nitrate production on sea salt and dust is important to include in models as it tends to dominate nitrate production and controls its partitioning between the fine and coarse mode. In that sense, it seems to me that 2 and 3 above are referring to the same process."

Author's response: "We intend to separate discussion of section 5.2 and 5.3 because the nitrate formation on the surface of sea salt and dust aerosol (section 5.3) is important, but not the only factor, to determining nitrate size distribution (section 5.2). Also, the former focuses more on chemical process and the later on physical process and climate implication." I find that the authors did not address the reviewer's concern. The formation of nitrate on coarse mode dust and sea salt particles is the major factor controlling the size distribution and the ratio of coarse/total particulate nitrate in models. This is not clearly explained in the manuscript, and should be clarified in section 5.

R1 comment #2: Authors did not address the reviewer's concern "What the authors are referring to by the use of "heterogeneous chemistry" is what I would call thermodynamic partitioning between the gas and aerosol phase." I agree with the reviewer that the use of the heterogeneous chemistry should not be applied to refer to the gas-particle thermodynamic partitioning. It is unclear in the revised manuscript whether the models are treating the formation of coarse mode nitrate by a heterogeneous uptake of HNO3 onto dust and sea-salt particles or by an equilibrium approach. This must be clarified.

**Concerning responses to R2:**

In the added text please change the 1st sentence to: "Our work presents an initial effort to assess nitrate simulation from chemical and physical processes (deposition)." You should be more specific by what chemical and physical processes you have looked at. And please have the rest of the paragraph corrected for English.

Why would you want to compromise the accuracy and efficiency, this sentence is misleading (line 619: "Several approximations, therefore, have been developed to compromise accuracy and efficiency.") Is this what you meant: "Several approximations have been developed to allow computational efficiency although they might compromise the model accuracy." Please reword.

**Additional comments:**

1) The confusion about the use of the "nitrate" term:

It should be clearly stated in the title that the paper is evaluating the particulate nitrate: "Investigation of global particulate nitrate from the AeroCom Phase III experiment."

Instead of using nitrate aerosol it would be preferable to use particulate nitrate when referring to the particle phase as aerosol term refers to both gas and particulate fraction that are in equilibrium.

Line 122: Is this particulate or gas-phase nitrate: "If fixed Nr is deposited as nitrate in forests,.."

2) As mentioned by R1, N2O5 hydrolysis is an important heterogeneous reaction when investigating the nitrate budgets that is typically included in global models. It should be clearly stated in Table 1 or 2 and in the paper how this reaction is treated and if it is included. And some discussion on the uncertainty due to this reaction and references should be added in the manuscript.

3) Add "relative" Line 78: More importantly, the relative importance of aerosol nitrate

4) Add "particulate" Line 102:  First, the formation of particulate nitrate,

5) Given that coarse mode nitrate measurements are sparse, please include measurements that have been done in Paris during the ESQUIF campaign that found that the coarse nitrate fraction represents up to 60% of the total particulate nitrate mass during the night and 80% during the day. See either Figure 13c of "Hodzic et al., ACP 2006 Aerosol chemical and optical properties over the Paris area within ESQUIF project", or Figure 6 of Hodzic et al, AE 2006: A model evaluation of coarse-mode nitrate heterogeneous formation on dust particles."

6) Clarify what you mean by feedback in this sentence: Line 202: "All models use full gas phase O3-NOx-HOx chemistry to produce HNO3 and consider the feedback of nitrate aerosol formation on HNO3 calculation." Do you mean radiative feedbacks on photolysis or changes in the HNO3 concentrations due to the gas/particle equilibrium?

---

## Author Response (AR2)

Thanks to co-Editor for your review. Here is our point-to-point reply.

**Concerning responses to R1:**

R1 comment #1: "They attribute these model differences to differences in 1) pH-dependent wet deposition of NH4+, 2) nitrate formation on the surface of sea salt and dust aerosol, and 3) the nitrate coarse mode fraction. They find that nitrate production on sea salt and dust is important to include in models as it tends to dominate nitrate production and controls its partitioning between the fine and coarse mode. In that sense, it seems to me that 2 and 3 above are referring to the same process."

Author's response: "We intend to separate discussion of section 5.2 and 5.3 because the nitrate formation on the surface of sea salt and dust aerosol (section 5.3) is important, but not the only factor, to determining nitrate size distribution (section 5.2). Also, the former focuses more on chemical process and the later on physical process and climate implication." I find that the authors did not address the reviewer's concern. The formation of nitrate on coarse mode dust and sea salt particles is the major factor controlling the size distribution and the ratio of coarse/total particulate nitrate in models. This is not clearly explained in the manuscript, and should be clarified in section 5.

**Authors' response: we added on lines 691-695 "Coarse mode aerosol nitrate is formed due to presence of dust and/or sea salt. In other words, the formation of nitrate on coarse mode dust and sea salt particles is the major factor controlling the size distribution. Other factors, such as $NH_3/NH_4^+/NO_3^-$ chemistry and atmospheric transport and removal processes, also affect nitrate size distribution."**

R1 comment #2: Authors did not address the reviewer's concern "What the authors are referring to by the use of "heterogeneous chemistry" is what I would call thermodynamic partitioning between the gas and aerosol phase." I agree with the reviewer that the use of the heterogeneous chemistry should not be applied to refer to the gas-particle thermodynamic partitioning. It is unclear in the revised manuscript whether the models are treating the formation of coarse mode nitrate by a heterogeneous uptake of HNO3 onto dust and sea-salt particles or by an equilibrium approach. This must be clarified.

**Authors' response: On lines 225-227, we state "Please note that the heterogeneous chemical production of particulate nitrate mentioned in this paper refers only to the first order loss reaction of $HNO_3$ on the surface of dust and sea salt particles." On lines 279-285, we indicate that there are two ways to account for the contribution of dust and sea salt to nitrate formation. Some models (EMAC, Oslo-CTM3, and Oslo-CTM2) include dust and/or sea salt components in their TEQM models directly (marked as TEQM in table 1 under column "How do CHEMDUSS"), while some models (EMEP, GISS-OMA, GMI, and INCA) use an approach of first order loss rate outside their TEQMs to account for the heterogeneous reactions of $HNO_3$ on the surface of dust and sea salt (marked as HETCHEM in table 1). We have gone through the paper to clarify "heterogeneous chemistry", see lines 34-35, 222, 226-227, and 826.**

**Concerning responses to R2:**

In the added text please change the 1st sentence to: "Our work presents an initial effort to assess nitrate simulation from chemical and physical processes (deposition)." You should be more specific by what chemical and physical processes you have looked at. And please have the rest of the paragraph corrected for English.

**Author's response: Changed the sentence to "Our work presents a first effort to assess nitrate simulation from chemical (e.g. chemistry among $NH_3$, $NH_4^+$, $NO_3^-$, $SO_4^{2-}$, dust and sea salt) and physical processes (e.g emission, dry deposition, and wet deposition)." The whole paragraph has been revised for English.**

Why would you want to compromise the accuracy and efficiency, this sentence is misleading (line 619: "Several approximations, therefore, have been developed to compromise accuracy and efficiency.") Is this what you meant: "Several approximations have been developed to allow computational efficiency although they might compromise the model accuracy." Please reword.

**Author's response: Done as suggested.**

**Additional comments:**

1) The confusion about the use of the "nitrate" term:

It should be clearly stated in the title that the paper is evaluating the particulate nitrate: "Investigation of global particulate nitrate from the AeroCom Phase III experiment."

Instead of using nitrate aerosol it would be preferable to use particulate nitrate when referring to the particle phase as aerosol term refers to both gas and particulate fraction that are in equilibrium.

**Author's response: Done as suggested.**

Line 122: Is this particulate or gas-phase nitrate: "If fixed Nr is deposited as nitrate in forests,.."

**Authors' response: Nr (Reactive nitrogen) is a term used for a variety of nitrogen compounds that support growth directly or indirectly. Nr includes the gases nitrogen oxides ($NO_x$), ammonia ($NH_3$), nitrous oxide ($N_2O$), as well as gas and particulate nitrate ($NO_3^-$). I added an explanation "including gas and particulate $NO_3^-$ and other nitrogen compounds" in lines 124-125.**

2) As mentioned by R1, N2O5 hydrolysis is an important heterogeneous reaction when investigating the nitrate budgets that is typically included in global models. It should be clearly stated in Table 1 or 2 and in the paper how this reaction is treated and if it is included. And some discussion on the uncertainty due to this reaction and references should be added in the manuscript.

**Authors' response: We added a column "N2O5 hydrolysis" in table 1. We also added these sentences in lines 209-215.**

**"Meanwhile, all models consider $N_2O_5$ hydrolysis, the conversion of $N_2O_5$ to $HNO_3$. The first order loss reaction occurs on the surface of tropospheric aerosols and assumes irreversible instant reaction. However, the models differ in $N_2O_5$ hydrolysis by considering the reaction on the surface of different aerosol types. Uptake coefficients (aka gamma factors) also differ in their relationship to temperature and RH. CHASER model is special as it allows $N_2O_5$ conversion to $HNO_3$ on liquid cloud particles. Please refer to Table 1 and the listed references for details."**

3) Add "relative" Line 78: More importantly, the relative importance of aerosol nitrate

**Authors' response: Done.**

4) Add "particulate" Line 102: First, the formation of particulate nitrate,
**Authors' response: Done.**

5) Given that coarse mode nitrate measurements are sparse, please include measurements that have been done in Paris during the ESQUIF campaign that found that the coarse nitrate fraction represents up to 60% of total particulate nitrate mass during the night and 80% during the day. See either Figure 13c of "Hodzic et al., ACP 2006 Aerosol chemical and optical properties over the Paris area within ESQUIF project", or Figure 6 of Hodzic et al, AE 2006: A model evaluation of coarse-mode nitrate heterogeneous formation on dust particles."
**Authors' response: A sentence has been added on lines 746-748: "Measurements taken in Paris during the ESQUIF campaign found that the coarse nitrate fraction represents up to 60% of the total particulate nitrate mass at night and 80% at day (Hodzic et al., 2006a, 2006b)."**

6) Clarify what you mean by feedback in this sentence: Line 202: "All models use full gas phase O3-NOx-HOx chemistry to produce HNO3 and consider the feedback of nitrate aerosol formation on HNO3 calculation." Do you mean radiative feedbacks on photolysis or changes in the HNO3 concentrations due to the gas/particle equilibrium?
**Authors' response: We added an explanation on lines 208-209: "changes in HNO3 concentrations due to the gas/particle equilibrium)."**

[revised manuscript text omitted]

concentrations due to the gas/particle equilibrium). Meanwhile, all models consider $N_2O_5$
hydrolysis, the conversion of $N_2O_5$ to $HNO_3$. The first order loss reaction occurs on the
surface of tropospheric aerosols and assumes irreversible instant reaction. However, the
models differ in $N_2O_5$ hydrolysis by considering the reaction on the surface of different
aerosol types. Uptake coefficients (aka gamma factors) also differ in their relationship to
temperature and RH. CHASER model is special as it allows $N_2O_5$ conversion to $HNO_3$
on liquid cloud particles. Please refer to Table 1 and the listed references for details.

[revised manuscript text omitted]

mode aerosol nitrate is formed due to presence of dust and/or sea salt. In other words, the
formation of nitrate on coarse mode dust and sea salt particles is the major factor
controlling size distribution. Other factors, such as $NH_3/NH_4^+/NO_3^-$ chemistry and
atmospheric transport and removal processes, also affect nitrate size distribution. Having
an accurate aerosol size distribution is critical in climate forcing estimations, since large
size particles have a relatively small optical cross section at a given aerosol mass loading
and the nitrate material coating on dust particles has almost no direct impact on the dust
optics, although the greatly impact dust lifetime (Bauer et al., 2007). Given that the
deposition velocity of a coarse particle is greater than that of a fine particle, an accurate
size distribution is also necessary to estimate deposition of particulate nitrates (Yeatman
et al., 2001; Sadanaga et al., 2008). This estimation is particularly important over oceans
where coarse mode nitrate dominates (Itahashi et al., 2016) and nitrogen supply is often
in deficit (Hansell and Follows, 2008).

As we have discussed in section 5.2, nitrate size distribution varies with the approaches
adopted for nitrate formation on coarse mode aerosols (i.e. dust and sea salt). Figure 10
gives the burdens of nitrate in fine mode and coarse mode portions and the ratio between
coarse mode and total (f_c) for the eight discussed models. The ratio is ranging from 0
(CHASER and GISS-OMA), ~50% (EMAC, GMI and INCA), ~80% (EMEP and
OsloCTM2), and 97% (OsloCTM3). The two OsloCTMs give the highest f_c partially
because they run TEQM model for coarse model particles.

A wide range of f_c, from 0 to > 90%, has been reported previously by model simulations
(Adams et al., 2001; Bauer et al., 2007; Jacobson 2001), while the range is narrowed
down to 40-60% for the model studies using the approach that solves dynamic mass
transfer equation for coarse mode particles (Feng and Penner, 2007; Xu and Penner,
2012).

It is worth pointing out that aerosol microphysics modify aerosol size as well. For
example, a process like coagulation would also allow $NO_3^-$ to mix with other particles and
enter coarse mode aerosol. New particle formation/nucleation would add $NH_3/NH_4^+/NO_3^-$
into the ultra fine mode. Except EMAC and GISS-MATRIX, majority models involved in
this study are bulk aerosol models that do not account for aerosol microphysics.

It is challenging to verify the nitrate size distribution globally due to the limited
measurements on time and space. Measurements over regional and station sites indicated
that the ratio of f_c could be very high and vary seasonally over oceanic sites. For
example, annual mean f_c during 2002-2004 from the Fukue supersite observatory is
about 72% with a seasonal variation of 60–80% in winter and of around 80% in summer
(Itahashi et al., 2016).

However, the ratio could be varied dramatically over land or the areas affected by land
pollution. For example, observations of fine and coarse particulate nitrate at several rural
locations in the United States indicated that nitrate was predominantly in submicron
ammonium nitrate particles during the Bondville and San Gorgonio (April) campaigns, in coarse mode nitrate particles at Grand Canyon (May) and Great Smoky Mountains
(July/August), and both fine and coarse mode nitrate during the studies at Brigantine and
San Gorgonio (July) (Lee et al., 2008). Allen et al. (2015) examined aerosol composition
data collected during the summer 2013 SOAS and concluded that inorganic nitrate in the
southeastern United States likely exists in the form of supermicron $NO_3^-$, balanced by the
presence of mineral cations arising from the transport of crustal dust and sea spray
aerosol. The measurements over Harvard Forest, a rural site in central Massachusetts,
supported that the majority of nitrate mass was associated with water-soluble
supermicron soil-derived $Ca^{2+}$ in an acidic environment (Lefer and Talbot, 2001).
Measurements taken in Paris during the ESQUIF campaign found that the coarse
nitrate fraction represents up to 60% of total particulate nitrate mass at night and 80% at

[revised manuscript text omitted]
E., Diskin, G. S., Fisher, J. A., Fuelberg, H. E., Hecobian, A., Knapp, D. J., Mikoviny, T., Riemer, D.,
Sachse, G. W., Sessions, W., Weber, R. J., Weinheimer, A. J., Wisthaler, A., and Jimenez, J. L.:
Effects of aging on organic aerosol from open biomass burning smoke in aircraft and laboratory
studies, Atmos. Chem. Phys., 11, 12049–12064, doi:10.5194/acp-11-12049-2011, 2011.

Davis, J. M., P. M. Bhave, and K. M. Foley (2008), Parameterization of N2O5 reaction probabilities on the
surface of particles containing ammonium, sulfate and nitrate, Atmos. Chem. Phys., 8, 5295 – 5311.

~~Cubison, M.J., A.M. Ortega, P.L. Hayes, D.K. Farmer, D.Day, M.J. Lechner, W.H. Brune, E. Apel, G.S.
Diskin, J.A. Fisher, H.E. Fuelberg, A. Hecobian, D.J. Knapp, T. Mikoviny, D. Riemer, G.W. Sachse,
W. Sessions, R.J. Weber, A.J. Weinheimer, A. Wisthaler, and J.L. Jimenez (2011), Effects of Aging on
Organic Aerosol from Open Biomass Burning Smoke in Aircraft & Lab Studies. Atmos. Chem. and
Phys. Disc. 11, 12103-12140, doi:10.5194/acpd-11-12103-2011.~~

DeCarlo, P. F., Kimmel, J. R., Trimborn, A., Northway, M. J., Jayne, J. T., Aiken, A. C., Gonin, M.,
Fuhrer, K., Horvath, T., Docherty, K. S., Worsnop, D. R., and Jimenez, J. L.: Field-deployable, high-
resolution, time-of-flight aerosol mass spectrometer, Anal. Chem., 78(24), 8281–8289, 2006.

Dentener, F. J., G. R. Carmichael, Y. Zhang, J. Lelieveld, and P. J. Crutzen, Role of mineral aerosol as a
reactive surface in the global troposphere, J. Geophys. Res, 101, 22,869-22889, 1996.

Dentener, F. and Crutzen, P.: Reaction of NO on Tropospheric Aerosols: Impact on the Global
Distributions of NO, O, and OH, J. Geophys. Res., 98, 7149–7163, doi:10.1029/92JD02979, 1993.

Dentener, F., Kinne, S., Bond, T., Boucher, O., Cofala, J., Generoso, S., Ginoux, P., Gong, S., Hoelzemann,
J. J., Ito, A., Marelli, L., Penner, J. E., Putaud, J.-P., Textor, C., Schulz, M., van der Werf, G. R., and
Wilson, J.: Emissions of primary aerosol and precursor gases in the years 2000 and 1750 prescribed
data-sets for AeroCom, Atmos. Chem. Phys., 6, 4321–4344, doi:10.5194/acp-6-4321-2006, 2006.

Evans, M. J. and Jacob, D. J.: Impact of new laboratory studies of N2O5 hydrolysis on global model
budgets of tropospheric nitrogen oxides, ozone and OH, Geophys. Res. Lett., 32, L09813,
doi:10.1029/2005GL022469, 2005.

[revised manuscript text omitted]